# Mechanisms influencing seasonal-to-interannual prediction skill of sea ice extent in the Arctic Ocean in MIROC

Jun Ono[1], Hiroaki Tatebe[1], Yoshiki Komuro[1], Masato I. Nodzu[2], Masayoshi Ishii[3]

[1] Japan Agency for Marine-Earth Science and Technology, Yokohama, 236-0001, Japan
[2] Tokyo Metropolitan University, Hachioji, 192-0397, Japan
[3] Meteorological Research Institute, Japan Meteorological Agency, Tsukuba, 305-0052, Japan

*Correspondence to*: Jun Ono (jun.ono@jamstec.go.jp)

**Abstract.** To assess the skill of seasonal-to-interannual predictions of the detrended sea ice extent in the Arctic Ocean ($SIE_{AO}$) and to clarify the underlying physical processes, we conducted ensemble hindcasts, started on January 1st, April 1st, July 1st, and October 1st for each year from 1980 to 2011, for lead times up to three years, using the Model for Interdisciplinary Research on Climate (MIROC) version 5 initialized with the observed atmosphere and ocean anomalies and sea ice concentration. Significant skill is found for the winter months: the December $SIE_{AO}$ can be predicted up to 11 months ahead (anomaly correlation coefficient is 0.42). This skill might be attributed to the subsurface ocean heat content originating in the North Atlantic. A plausible mechanism is as follows: the subsurface water flows into the Barents Sea from spring to fall and emerges at the surface in winter by vertical mixing, and eventually affects the sea ice variability there. Meanwhile, the September $SIE_{AO}$ predictions are skillful for lead times of up to 2 months, due to the persistence of sea ice in the Beaufort, Chukchi, and East Siberian Seas initialized in July, as suggested by previous studies.

## 1 Introduction

The Arctic has warmed more than twice as much as the global average (e.g., Bekryaev et al., 2010; Cohen et al., 2014), called Arctic amplification. Sea ice reduction under climate change is one of the main processes contributing to Arctic amplification (e.g., Pithan and Mauritsen, 2014). Arctic summer sea ice extent has declined at about 14 % per decade (National Snow and Ice Data Center, 2016, http://nsidc.org/arcticseaicenews/). In September 2012, sea ice extent reached its minimum since satellite observations began in the late 1970s. Moreover, Arctic sea ice thickness has decreased by around 65 % from 1975 to 2012 (Kwok et al., 2009, Lindsay and Schweiger, 2015).

In contrast to the rapid warming in the Arctic, severely cold winters have occurred more frequently at midlatitudes. Although the exact cause is still being debated (e.g., Barnes and Screen, 2015), Mori et al. (2014) have shown, using ensemble experiments with an atmospheric general circulation model, that the more frequent cold winters at midlatitudes can be partly explained by the sea ice decrease in the Barents and Kara Seas. Therefore, further investigation of the mechanisms driving Arctic sea ice variability is of great importance for more accurate projections of climate change, not only in the Arctic but also for the midlatitudes.

A previous study based on two and five years perfect-model experiments from January 1st and September 1st has shown that the potential predictability for sea ice extent remains statistically significant at lead times up to 1-2 years, primarily because of the persistence of ice thickness anomalies from summer to summer and the persistence of sea surface temperature anomalies from the melt to growth seasons (Blanchard-Wrigglesworth et al., 2011a; Guemas et al., 2014). These features are also found in the results of experiments comparing multiple climate models (Day et al., 2014b; Tietsche et al., 2014). The observed detrended Arctic sea ice extent based on ensemble hindcasts can be predicted up to 2–7 and 5–11 months ahead for summer and winter, respectively (e.g., Chevallier et al., 2013; Sigmond et al., 2013; Wang et al., 2013; Msadek et al., 2014; Peterson et al., 2015; Guemas et al., 2016; Sigmond et al., 2016). In these ensemble hindcasts, it is found that the ice thickness and the surface or subsurface water temperatures are closely related to the prediction skill, as suggested by idealized or perfect-model experiments with climate models (e.g., Blanchard-Wrigglesworth et al., 2011b; Chevallier and Salas-Mélia, 2012; Day et al., 2014a).

Until very recently, the mechanisms by which the above variables contribute to the prediction skill had not been quantified. Bushuk et al. (2017) examined the physical mechanisms underlying the prediction skill of regional sea ice extent and showed for the first time the importance of the initializations of ocean subsurface and sea ice thickness in their dynamical prediction system.

Motivated by the above studies, we first conduct initialized ensemble hindcasts using a climate model to assess the seasonal-to-interannual predictability of sea ice extent in the Arctic Ocean and further investigate sources for prediction skill and clarify the physical processes linking the prediction skill to its sources. In particular, the present study reveals that subsurface ocean heat content originating from the North Atlantic contributes to the predictability of winter sea ice through advection and vertical mixing processes, which is somewhat different from the reemergence process of the local subsurface ocean temperature suggested by Bushuk et al. (2017).

## 2 Experimental Designs

The climate model used here is a low-resolution version of the Model for Interdisciplinary Research on Climate, version 5 (MIROC5) (Watanabe et al., 2010), which contributed to the fifth phase of the Coupled Model Intercomparison Project and the Intergovernmental Panel on Climate Change Fifth Assessment Report (IPCC AR5, 2013). The atmospheric component has a horizontal resolution of T42 spectral truncation (approximately 2.8°) and comprises 40 vertical layers up to 3 hPa. The oceanic component has horizontal resolutions of 1.4° in longitude and 0.5–1.4° in latitude, and comprises 50 vertical layers. The sea ice component of MIROC5 contains one-layer thermodynamics (Bitz and Lipscomb, 1999), elastic-viscous-plastic rheology (Hunke and Dukowicz, 1997), and the subgrid ice thickness distribution (Bitz et al., 2001) with five categories; the detailed structure has been described in Komuro et al. (2012).

To initialize MIROC5, we adopted anomaly assimilation for the atmosphere and ocean and full-field assimilation for sea ice. Anomalies were calculated as the deviations from the climatology defined by the 1961–2000 period. The observed 6-hourly air temperature and wind vectors from the 55-year Japanese Reanalysis (JRA-55) dataset (Kobayashi et

al., 2015) were linearly interpolated to the atmospheric model's grid. The observed monthly ocean temperature, salinity, and sea ice concentration (SIC) from the gridded monthly objective analysis produced by Ishii et al. (2006) and Ishii and Kimoto (2009) were linearly interpolated to obtain the daily values, and the same grid as the ocean model. The ocean data are based on the latest observational databases (the World Ocean Database (WOD05), World Ocean Atlas (WOA05), and Global

Temperature Salinity Profile Program (GTSPP) provided by the U.S. National Oceanographic Data Center (NODC)) and a SST analysis (Centennial in situ Observation Based Estimates of variability of SST and marine meteorological variables (COBE SST); Ishii et al. (2005); Hirahara et al. (2014)). The SIC data are based on satellite observations from the Nimbus-5 Scanning Multichannel Microwave Radiometer (SMMR), the Special Sensor Microwave Imager (SSM/I), and the Special Sensor Microwave Imager/Sounder (SSMIS; Armstrong et al., 2012).

10        In the assimilation runs, the atmospheric anomalies were assimilated into MIROC5 below 100 hPa at 6-hourly intervals and the oceanic anomalies above 3000 m depth at one-day intervals except in sea ice regions, using a modified incremental analysis update scheme (Tatebe et al., 2012). Meanwhile, SIC was assimilated at one-day intervals following Lindsay and Zhang (2006) and Stark et al. (2008). These assimilations were conducted over the period 1975–2011 with eight ensemble members produced by perturbing the sea surface temperature based on the observational errors. The atmospheric

and oceanic initial states were obtained from a non-initialized twentieth-century run with historical natural and anthropogenic forcings.

On the basis of the assimilation runs, the hindcast experiments were integrated for 3 years from January 1st, 2 years and 9 months from April 1st, 2 years and 6 months from July 1st and 2 years and 3 months from October 1st for each year from 1980 to 2011. The initial states of the atmosphere and ocean were obtained from the corresponding assimilation runs.

In addition, a control run with MIROC version 5.2, which is a minor update of MIROC5, was used to interpret the physical processes contributing to the prediction skill in the hindcasts. This simulation was run with external forcings fixed at the year 2000 levels under a multi-model inter-comparison project (Day et al., 2016).

In Sect. 3 and Sect. 4, we analyze the detrended monthly anomalies to extract the internal variations with seasonal-to-interannual timescales. Here, the detrended components were calculated by subtracting monthly linear trends during

1980–2009 from the original monthly data, and anomalies are defined as deviations from the climatology from 1980–2009. Moreover, climate drifts in the hindcasts are removed according to the INTERNATIONAL CLIVAR PROJECT OFFICE (ICPO, 2011). Here, the climate drift $T_{drf}$ is estimated as follows: $T_{drf}(\tau) = \frac{1}{N}\sum_{k=1}^{N}(T_p^k(\tau) - T_a^k(\tau))$, where $k = 1, \cdots, N$ is the initial time; $\tau$ is the forecast lead time; $T$ is the monthly quantity of interest, for example, the temperature and sea ice concentration; and the subscripts $p$ and $a$ represent the ensemble averaged prediction and the corresponding assimilation,

respectively. As mentioned in Sect. 1, sea ice reduction in the Arctic Ocean, especially in the Barents and Kara Seas, could lead to extreme weather at midlatitudes, which may be related to the warming of the Arctic Ocean interior (e.g., Polyakov et al., 2012). To clearly interpret the physical mechanisms influencing sea ice extent in the Arctic Ocean (hereafter SIE$_{AO}$), SIE$_{AO}$ is defined from the cumulative area for all grid cells north of 65° N with SIC greater than 15 %. From this definition,

Baffin Bay and Hudson Bay are partially included in the domain, but the directions of the main currents are from the Arctic Ocean interior (shelves and basins) to Baffin Bay through the straits of the Canadian Archipelago (e.g., Aksenov et al., 2011). Thus, the direct impacts of Baffin Bay and Hudson Bay on the Arctic Ocean interior are considered to be small. Note that the results of this study are not directly comparable with other hindcast studies that focus on pan-Arctic SIE (e.g., Chevallier et al., 2013; Sigmond et al., 2013; Wang et al., 2013; Msadek et al., 2014; Peterson et al., 2015; Guemas et al., 2016; Sigmond et al., 2016), due to the choice of Arctic Ocean domain. For comparison, the results for the detrended sea ice extent anomaly in the Northern Hemisphere are shown in the supporting information.

**3 Predictability of Arctic Sea Ice Extent**

We first examine the potential predictability of $SIE_{AO}$ (Fig. 1), based on the lagged auto-correlation coefficients, which is the skill of the persistence forecast. The lagged correlations with the observations (Ishii et al., 2006; Ishii and Kimoto, 2009) decrease within the first few months for all of the start months, and those originating between January and June subsequently rise again in the winter (November through March). Significant skill in the control run is obtained for greater lead times than in the observations, which is consistent with previous studies (e.g., Blanchard-Wrigglesworth et al., 2011b; Day et al., 2014b). For the SIE in the Northern Hemisphere (Fig. S1a), the correlation patterns are similar to those in Day et al. (2014b), except for a lead time of one month for May which may be due to differences in the observational time period (Fig. S1d). However, the reemergence in winter is weaker than that for $SIE_{AO}$. This is because the winter $SIE_{AO}$ variability is dominated by changes in the Barents and GIN Seas, which have long persistence timescales relative to other regions of winter sea ice variability.

We next evaluate the $SIE_{AO}$ prediction skill (Figs. 2a and 2b), with the anomaly correlation coefficient (ACC) and the root-mean-square error (RMSE) between the detrended observations and the hindcasts (e.g., Wang et al., 2013). Here, the RMSE values are normalized by the standard deviation of each month. In the hindcasts started from July 1st, the ACC for September is statistically significant and exceeds that of the persistence forecast, suggesting that September $SIE_{AO}$ can be dynamically predicted from the previous July (ACC = 0.79). Although the significance of the ACC is borderline, the results suggest that September $SIE_{AO}$ is potentially predictable from April 1st (ACC = 0.37), which is consistent with the results of Peterson et al. (2015). The ACC is also significant for the winter $SIE_{AO}$, in particular for December, except for the hindcasts started from April 1st, indicating the potential use of dynamical forecasts up to 11 months ahead (ACC = 0.42). The RMSE values for the first several lead months are smaller than the standard deviation for all hindcasts. The time series of September $SIE_{AO}$ shows that both the assimilation and hindcasts capture the observed interannual variability, although the model underestimates the variability in the mid-1980s and mid-1990s (Fig. 2c). The observed $SIE_{AO}$ in December is contained within the ensemble spread, excluding the mid-1980s (Fig. 2d). We also show the same figure as Fig. 2 in Fig. S2, except that the detrended sea ice extent anomaly is calculated for the Northern Hemisphere. The lower ACC at short lead times for the hindcasts started from January and April (Fig. S2a) may be due to the lower ACC and higher RMSE for sea ice concentration in the Sea of Okhotsk, the Bering Sea, and the Labrador Sea (not shown). The RMSE values in winter are

large (Fig. S2b) compared to Fig. 2b because $SIE_{AO}$ does not include the area where sea ice variability is large. The difference between Fig. 2d and Fig. S2d is also due to the effect of the domain choice.

## 4 Possible Mechanisms for Prediction Skill

Focusing on both the hindcasts started from January 1st, in which the December $SIE_{AO}$ has high skill even at long lead times, and those started from July 1st, in which the September $SIE_{AO}$ is significant, we examine mechanisms for the prediction skill. Figure 3 shows the lagged cross-correlations between the $SIE_{AO}$ and the sea ice volume in the Arctic Ocean ($SIV_{AO}$) and those between $SIE_{AO}$ and ocean heat content in the Arctic Ocean ($OHC_{AO}$) for the control run and the hindcasts started from January and July. Here, the $SIV_{AO}$ is defined as the sum of the grid cell volumes obtained by multiplying the sea ice thickness (SIT) by the SIC and the area for grid cells with SIC greater than 15 % and the $OHC_{AO}$ is the vertically integrated temperature multiplied by the density and specific heat capacity of seawater from the surface to a depth of 200 m, in the same area as the $SIE_{AO}$.

The $SIV_{AO}$ has stronger positive correlations with the $SIE_{AO}$ in summer than in winter (Figs. 3a–c), which is consistent with Chevallier and Salas-Mélia (2012), while the $OHC_{AO}$ has more persistent negative correlations with the $SIE_{AO}$ in winter than in summer (Figs. 3d–f). In the hindcasts started from January 1st, the December $SIE_{AO}$ is significantly correlated with the $OHC_{AO}$ from January to December. Similar features can be seen in the hindcasts started from July 1st. The $SIE_{AO}$ in September is significantly correlated with the $SIV_{AO}$ in July for both of the hindcasts, but only weakly correlated with the $OHC_{AO}$. Thus, sources for the prediction skill of the December and September $SIE_{AO}$ are suggested to be the ocean heat content from the surface to a depth of 200 m after January and the sea ice states in July, respectively. For the sea ice extent anomaly calculated in the Northern Hemisphere (Fig. S3), the patterns of the lagged correlation coefficients are broadly similar to those in Fig. 3. However, the correlations between the SIE and SIV are higher than those in the Arctic domain north of 65° N. One reason might be the contribution of sea ice variability south of 65° N. In addition, the correlations between SIE and OHC show weak positive values from June to October in the hindcasts. This is partly because the OHC includes the regions where sea ice does not exist throughout the year.

We next clarify the physical processes linking the prediction skill to sources of that skill. Figure 4 shows the SIC, SIT, and OHC north of 60° N regressed on the model-predicted December $SIE_{AO}$. The most significant signals for both SIC and SIT are found in the Barents Sea (BS) of the Arctic Ocean (Figs. 4a and 4b). It is well known that winter sea ice variability in the BS dominates that in the Arctic Ocean (e.g., Smedsrud et al., 2013), which is consistent with our results. At a lag of 9 months (Fig. 4c), negative correlation and regression coefficients for the OHC are found in regions from the northern part of the GIN Sea to the western part of the BS. The signals become strong in the western part of the BS at a lag of 6 months (Fig. 4d), and further extend across the entire BS at a lag of 3 months (Fig. 4e) and still appear in the BS at a lag of zero (Fig. 4f). These features are also found in the control run (Fig. S4), suggesting that the physical processes in the hindcasts are not due to processes distorted by the influence of initialization or climate drift in MIROC5. In contrast, the December $SIE_{AO}$ cannot be predicted from April 1st (Fig. 2a), although significant regression and correlation coefficients

appear in the results for the April hindcasts (Fig. S5). This may be because the RMSE for April SIC in the BS is larger in the April hindcasts than the January hindcasts (not shown). In this study, since we do not assimilate ocean data beneath the sea ice, initialized ocean states underneath the sea ice are considered to be different from the real ocean. Particularly, in the BS where sea ice variability is related to the skillful prediction of December $SIE_{AO}$, standard deviation of sea ice is larger in April than in January, and thus the initial shock might be large in April.

Considering that the Norwegian Atlantic Current tends to flow into the BS (e.g., Polyakov et al., 2005), the North Atlantic might be the source of the OHC anomaly contributing to the significant skill of the December $SIE_{AO}$. A plausible mechanism is as follows: the OHC anomalies initialized in the North Atlantic flow into the BS through advection, subsequently emerge at the surface due to vertical mixing in winter, and affect the December sea ice distribution in the BS and eventually in the Arctic Ocean. This hypothesis is partly supported by Nakanowatari et al. (2014). As originally proposed by Bushunk et al. (2017), our results suggest that the initialization of subsurface ocean temperature contributes to the skillful prediction of the winter sea ice extent in the BS.

For September, the sea ice states initialized in July persist until September in the Beaufort, Chukchi, and East Siberian Seas (Fig. 5), which is consistent with Bushuk et al. (2017). Consequently, this persistence contributes to the prediction skill of the September $SIE_{AO}$. In the hindcasts started from April 1st, the September $SIE_{AO}$ shows similar lagged correlation patterns to the July hindcasts for $SIV_{AO}$ (Fig. S6a) and $OHC_{AO}$ (Fig. S6b). Thus, the same physical processes as the July hindcasts are expected to be present in the April hindcasts. However, the positive regression and correlation patterns for SIC and SIT are lower than those for the July hindcasts, particularly in the Pacific Sector of the Arctic Ocean (Figs. S6c and S6d). In contrast, similar patterns to Fig. 5 clearly appear in the Pacific sector of the Arctic Ocean for the control experiment (Fig. S7). These results suggest that the persistence of sea ice contributes to the skill of September $SIE_{AO}$ started from April 1st, but errors in the initial conditions for SIT and model drift may lead to unclear signals in Fig. S6.

## 5 Concluding Remarks

We investigated the predictability of the detrended $SIE_{AO}$ anomaly and its sources based on an ensemble of hindcasts using an initialized climate model, MIROC5, and further identified physical processes related to the prediction skill. Prediction skill for Arctic winter $SIE_{AO}$ is significantly higher than the persistence forecast, especially for December, indicating the possibility for dynamical forecasting 11 months ahead. The December $SIE_{AO}$ is significantly correlated with the December SIC and SIT in the BS where the subsurface OHC anomalies might be advected from the North Atlantic, and subsequently emerge at the surface in winter, and contribute to the sea ice variability there. Our results suggest that sources of the December $SIE_{AO}$ prediction skill exist in the North Atlantic and thus initialization of the subsurface water there leads to better prediction of the $SIE_{AO}$ in December. Numerical experiments to confirm whether the subsurface OHC anomalies originating from the North Atlantic control the December sea ice extent in the BS and eventually in the Arctic Ocean will be explored in future work.

Significant skill for the September $SIE_{AO}$ is seen only up to 2 months ahead. Improvement in the prediction skill for summer $SIE_{AO}$ is dependent upon refinement of the initial state of the SIT. In fact, higher lagged correlations between the summer $SIE_{AO}$ and the $SIV_{AO}$ suggest that the initialization of the SIT is important, which is consistent with previous results by Day et al. (2014a) and Bushuk et al. (2017).

5          In recent years, the rapid reduction in Arctic sea ice has enabled ships to navigate the Northern Sea Route (e.g., Stephenson et al., 2014). Under such maritime activities in the Arctic Ocean, forecasts of the local sea ice distribution rather than the total sea ice extent become of greater interest for marine users. Recent studies have reported the forecast skills of the retreat and advance dates of the sea ice distribution based on statistical methods (e.g., Stroeve et al., 2016; Wang et al., 2016) as well as a dynamical forecast system (Sigmond et al., 2016; Bushuk et al., 2017). In the present study, our hindcasts could

not reproduce precise sea-ice edges from summer to fall. For example, the predicted sea ice distributions in September 2007 are overestimated in the Russian region of the Arctic Ocean. This is because the surface winds, which are thought to be the major driving force of sea ice motion in September 2007, are not adequately predicted. Other reasons might be the lower resolution of the ocean model or bias in the climatology. Further improvements in the skill to predict sea ice, including its spatial pattern, will be provided by climate models with higher resolution, reduced model drift and bias, and improved

initialization techniques.

*Data availability.* The data for this paper can be accessed via the authors for research purposes.

*Competing interests.* The authors declare that they have no conflict of interest.


*Acknowledgements.* This work was supported by the Program for Generation of Climate Change Risk Information (SOUSEI project) and the Arctic Challenge for Sustainability Project (ArCS Project), of the Japanese Ministry of Education, Culture, Sports, Science and Technology. J.O. was supported by Japan Society for the Promotion of Science (JSPS) Grant-in-Aid for Young Scientists (B) 17K12830. Numerical experiments were conducted on the Earth Simulator at the Japan Agency for

Marine-Earth Science and Technology. We also thank Takashi Mochizuki for his helpful discussions.

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

**Figure captions**

**Figure 1**: Lagged auto-correlation coefficients of the detrended $SIE_{AO}$ anomaly derived from (a) observations (Ishii et al., 2006; Ishii and Kimoto, 2009) and (b) a model control simulation, for each start month, against lead time, following Day et al. (2014b). Solid and dashed lines denote values for the September and March target months, respectively. Black dots indicate statistical significance at the 95 % confidence level based on a two-sided Student's $t$-test with 30 and 200 degrees of freedom in the observations and model, respectively.

**Figure 2**: Lead time dependence of (a) $SIE_{AO}$ ACC and (b) $SIE_{AO}$ RMSE ($\times 10^6$ km$^2$) for January, April, July, and October start hindcasts. The $SIE_{AO}$ ACC scores of hindcasts, which are higher than those of the persistence forecast and statistically significant at the 95 % confidence level based on a two-sided Student's $t$-test, are denoted by black dots. The $SIE_{AO}$ RMSE scores, which are normalized by the standard deviation, are denoted by black dots if the values are less than 1.0. Boxes in (a) indicate the lead time of the time series shown in (c) and (d). Time series of the detrended $SIE_{AO}$ anomaly for (c) September and (d) December, from the observations (OBSE; black line), assimilation (ASSI; red line), and hindcasts started from July 1st and January 1st (HIND.JUL and HIND.JAN; blue line). HIND.JUL is the September $SIE_{AO}$ at 2 months lead time and HIND.JAL is the December $SIE_{AO}$ at 11 months lead time. Blue shading indicates the ensemble spread. In (c), the September $SIE_{AO}$ at 5 months lead time started from April 1st (HIND.APR) is superimposed by an aqua line and shading.

**Figure 3**: Lagged correlation coefficients between the detrended $SIE_{AO}$ anomaly and (a–c) the detrended $SIV_{AO}$ anomaly and (d–f) the detrended $OHC_{AO}$ anomaly. Left, middle, and right panels indicate values obtained from the control run (CTRL), the hindcasts started from January 1st (HIND.JAN), and the hindcasts started from July 1st (HIND.JUL), respectively. Black dots indicate statistical significance at the 95 % confidence level based on a two-sided Student's $t$-test with 30 and 200 degrees of freedom in the observations and model. Note that the horizontal and vertical axes in the hindcasts started from July 1st are different from those in the control run and the hindcasts started from January 1st.

**Figure 4**: Lagged correlation (colors) and regression (contours) coefficients between the $SIE_{AO}$ anomaly ($\times 10^6$ km$^2$) in December and (a) SIC anomaly (%) at a lag of 0 months, (b) SIT anomaly (cm) at a lag of 0 months, and OHC anomalies ($\times 10^{18}$ J) at lags of (c) $-9$, (d) $-6$, (e) $-3$, and (f) 0 months, in regions from 60° to 90° N on the basis of the hindcasts started from January 1st. Contours are drawn at intervals of 5 (%) from 5 to 25 for SIC and 10 (cm) from 10 to 40 for SIT. In (c–f), the contours are drawn from $-1.0$ to $-0.1$ ($\times 10^{18}$ J) at intervals of 0.1 ($\times 10^{18}$ J). Stippling indicates regions with statistically significant correlation coefficients at the 95 % confidence level. White shading indicates areas where sea ice does not exist. A latitude circle of 65° N is also indicated by a thin solid line.

**Figure 5**. Lagged correlation (colors) and regression (contours) coefficients between the September $SIE_{AO}$ anomaly ($\times 10^6$ km$^2$) and (a) SIC anomaly (%) and (b) SIT anomaly (cm), based on the hindcasts started from July 1st. Contours are drawn at intervals of 5 (%) from 5 to 20 and at intervals of 10 (cm) from 10 to 40 for the SIC and SIT anomalies, respectively. Stippling indicates regions with statistically significant correlation coefficients at the 95 % confidence level. White shading indicates areas where sea ice does not exist. A latitude circle of 65° N is also indicated by a thin solid line.

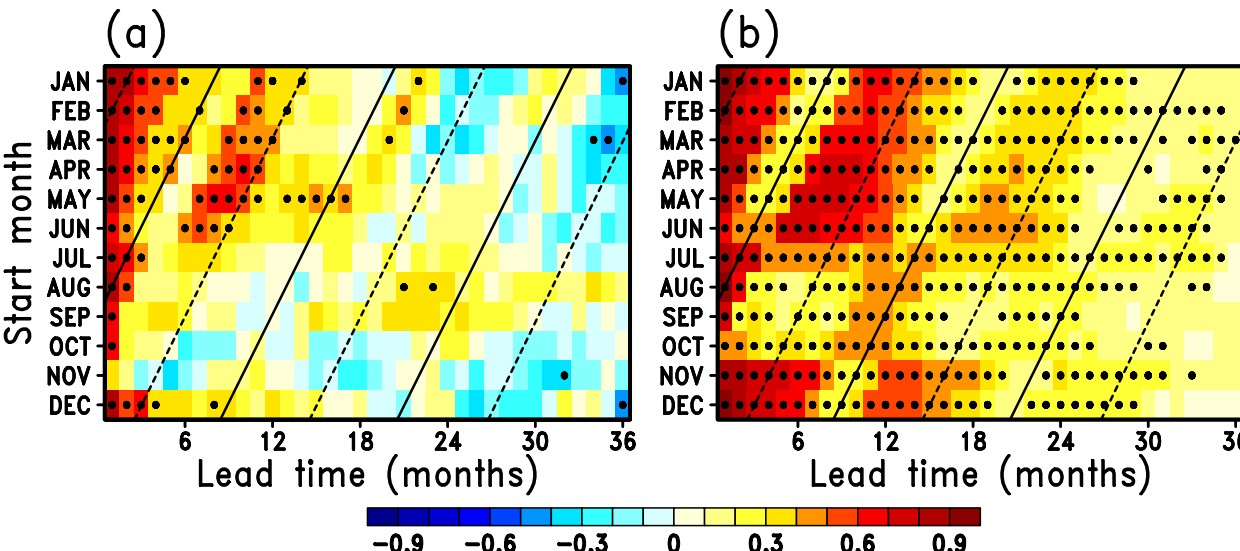

**Figure 1. Lagged auto-correlation coefficients of the detrended SIE$_{AO}$ anomaly derived from (a) observations (Ishii et al., 2006; Ishii and Kimoto, 2009) and (b) a model control simulation, for each start month, against lead time, following Day et al. (2014b). Solid and dashed lines denote values for the September and March target months, respectively. Black dots indicate statistical significance at the 95 % confidence level based on a two-sided Student's *t*-test with 30 and 200 degrees of freedom in the observations and model, respectively.**

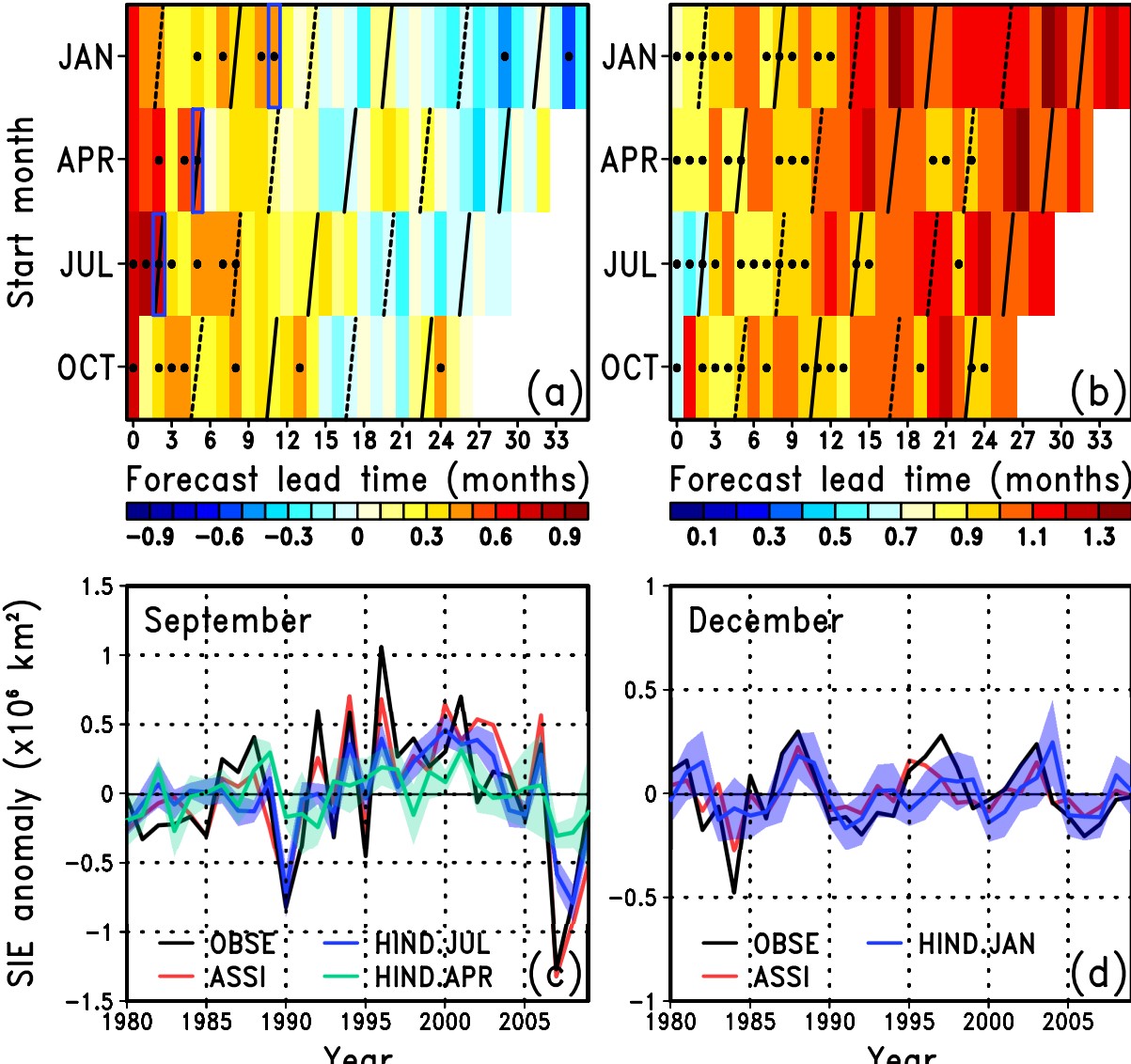

**Figure 2.** Lead time dependence of (a) $SIE_{AO}$ ACC and (b) $SIE_{AO}$ RMSE ($\times 10^6$ km$^2$) for January, April, July, and October start hindcasts. The $SIE_{AO}$ ACC scores of hindcasts, which are higher than those of the persistence forecast and statistically significant at the 95 % confidence level based on a two-sided Student's *t*-test, are denoted by black dots. The $SIE_{AO}$ RMSE scores, which are normalized by the standard deviation, are denoted by black dots if the values are less than 1.0. Boxes in (a) indicate the lead time of the time series shown in (c) and (d). Time series of the detrended $SIE_{AO}$ anomaly for (c) September and (d) December, from the observations (OBSE; black line), assimilation (ASSI; red line), and hindcasts started from July 1st and January 1st (HIND.JUL and HIND.JAN; blue line). HIND.JUL is the September $SIE_{AO}$ at 2 months lead time and HIND.JAL is the December $SIE_{AO}$ at 11 months lead time. Blue shading indicates the ensemble spread. In (c), the September $SIE_{AO}$ at 5 months lead time started from April 1st (HIND.APR) is superimposed by an aqua line and shading.

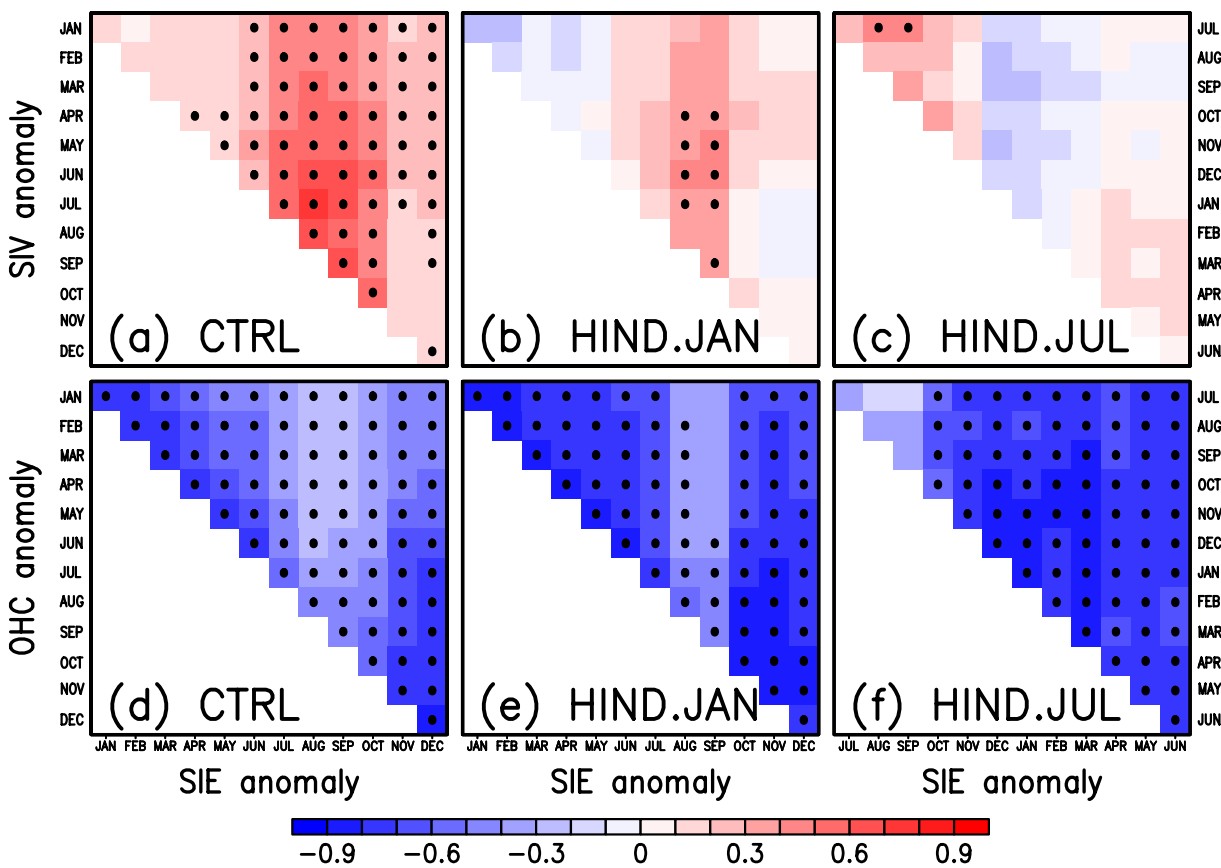

**Figure 3: Lagged correlation coefficients between the detrended SIE$_{AO}$ anomaly and (a–c) the detrended SIV$_{AO}$ anomaly and (d–f) the detrended OHC$_{AO}$ anomaly. Left, middle, and right panels indicate values obtained from the control run (CTRL), the hindcasts started from January 1st (HIND.JAN), and the hindcasts started from July 1st (HIND.JUL), respectively. Black dots indicate statistical significance at the 95 % confidence level based on a two-sided Student's *t*-test with 30 and 200 degrees of freedom in the observations and model. Note that the horizontal and vertical axes in the hindcasts started from July 1st are different from those in the control run and the hindcasts started from January 1st.**

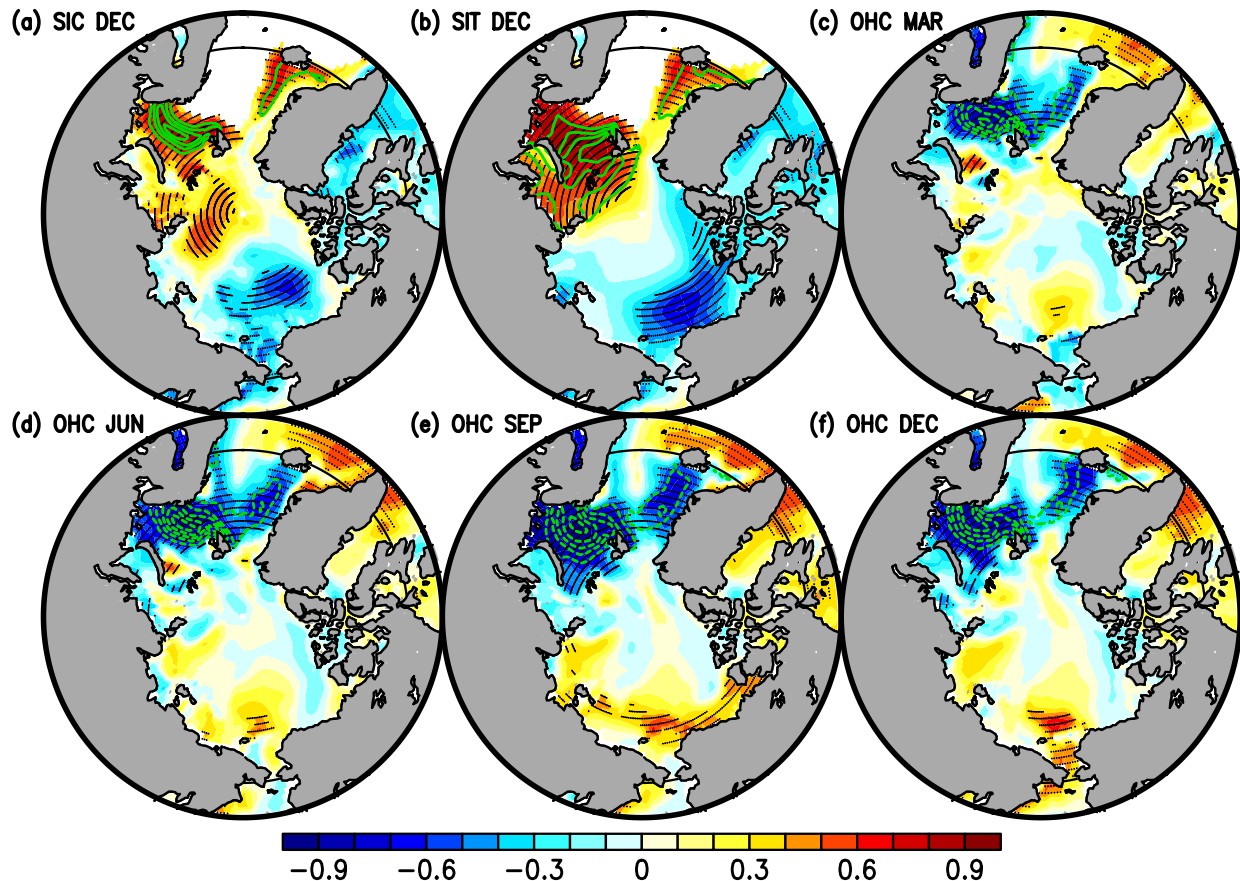

**Figure 4.** Lagged correlation (colors) and regression (contours) coefficients between the $SIE_{AO}$ anomaly ($\times 10^6$ km$^2$) in December and (a) SIC anomaly (%) at a lag of 0 months, (b) SIT anomaly (cm) at a lag of 0 months, and OHC anomalies ($\times 10^{18}$ J) at lags of (c) $-9$, (d) $-6$, (e) $-3$, and (f) 0 months, in regions from 60° to 90° N on the basis of the hindcasts started from January 1st. Contours are drawn at intervals of 5 (%) from 5 to 25 for SIC and 10 (cm) from 10 to 40 for SIT. In (c–f), the contours are drawn from $-1.0$ to $-0.1$ ($\times 10^{18}$ J) at intervals of 0.1 ($\times 10^{18}$ J). Stippling indicates regions with statistically significant correlation coefficients at the 95 % confidence level. White shading indicates areas where sea ice does not exist. A latitude circle of 65° N is also indicated by a thin solid line.

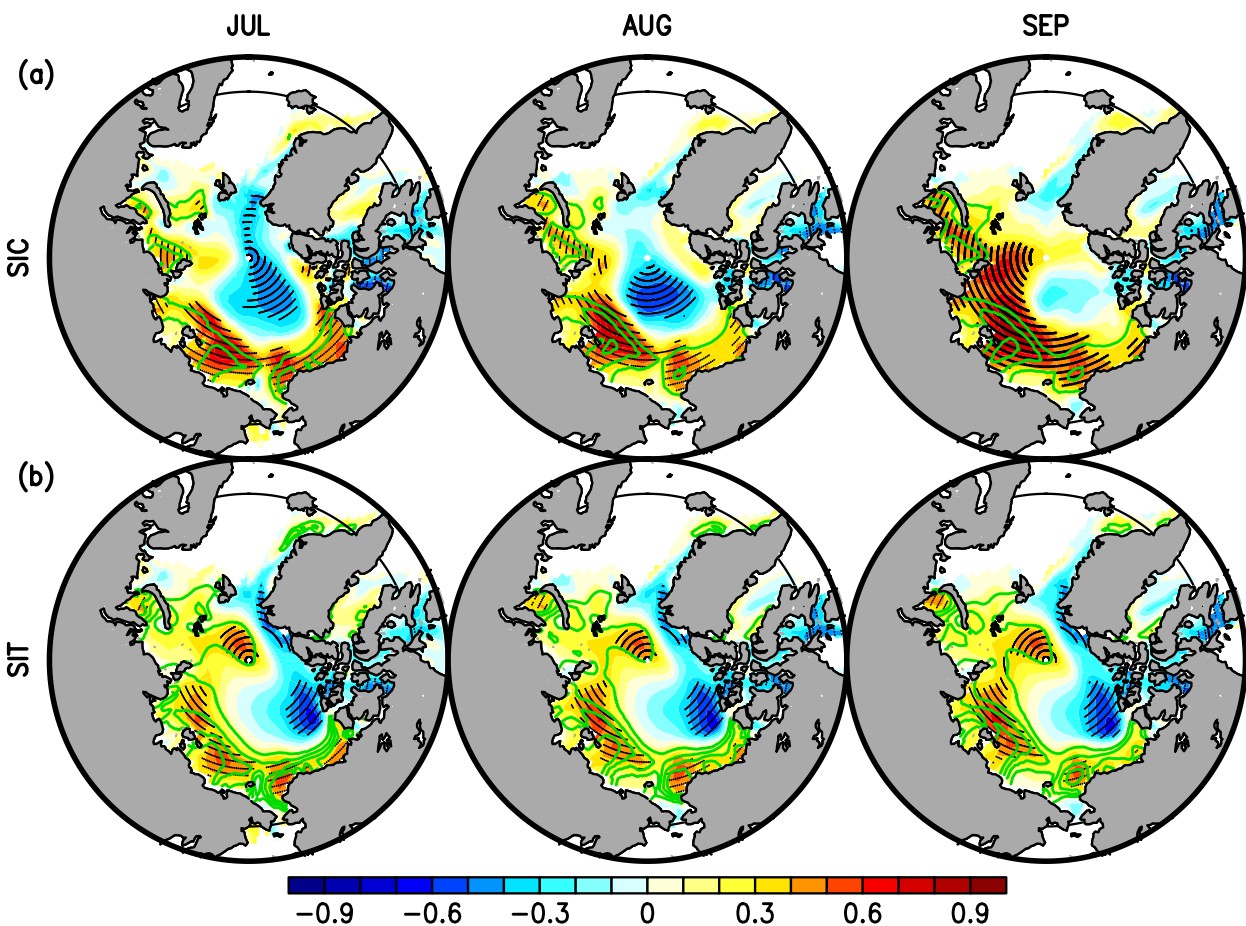

Figure 5. Lagged correlation (colors) and regression (contours) coefficients between the September SIE$_{AO}$ anomaly ($\times 10^6$ km$^2$) and (a) SIC anomaly (%) and (b) SIT anomaly (cm), based on the hindcasts started from July 1st. Contours are drawn at intervals of 5 (%) from 5 to 20 and at intervals of 10 (cm) from 10 to 40 for the SIC and SIT anomalies, respectively. Stippling indicates regions with statistically significant correlation coefficients at the 95 % confidence level. White shading indicates areas where sea ice does not exist. A latitude circle of 65° N is also indicated by a thin solid line.