# Peer review of "Mechanisms influencing seasonal-to-interannual prediction skill of sea ice extent in the Arctic Ocean in MIROC"

_The Cryosphere, 2017_

## Referee Comment (RC1) · Anonymous Referee #1 · 18 Aug 2017

This work investigates the seasonal-to-interannual prediction skill of Arctic sea-ice extent (SIE) using a set of hindcast experiments performed with the MIROC GCM. The authors investigate prediction skill for detrended Arctic SIE, identifying skillful predictions up to one year in advance. They also examine the key physical mechanisms impacting prediction skill, concluding that North Atlantic ocean heat content anomalies are a source of skill for December SIE predictions and that sea ice volume is a source of skill of September SIE predictions.

I commend the authors for their focus on physical mechanisms and their relation to the reported SIE prediction skill. However, I have a number of serious concerns with the

manuscript in its present form. In particular, my major concerns are: (1) the authors' choice of Arctic domain, and how this choice biases and confuses results throughout the manuscript; (2) the definition of ocean heat content and its impact on the proposed advective ocean heat content mechanism; and (3) the apparent disagreement of SIE lagged correlation values with previously published literature. Specific comments detailing these concerns are provided below.

Note: I will use the convention p.l throughout this review to refer to page number p and line number l of the discussion paper.

Major Comments:

Before beginning the major comments, I would like to clarify a convention. The authors use a different lead-naming convention than the hindcast studies cited on 2.7. For example, a July 1 forecast of September SIE is referred to as a "lead-2" forecast in the literature cited on 2.7. In the manuscript, the authors refer to this forecast as a "lead-3" forecast. The authors should change their naming convention to be consistent with previous hindcast studies. I will use the commonly used convention in this review.

Major Comment 1) Choice of Arctic domain

The author's define their Arctic Ocean domain as all gridpoints north of 65N. They also exclude Baffin Bay and Hudson Bay from their Arctic Ocean domain without providing any justification for this decision. The Arctic Ocean domain choice directly affects the interpretation of essentially all reported results in the paper. I suspect that Figures 1, 2, 3, and 4 would all be notably different if the authors analyzed the commonly used pan-Arctic domain (i.e. all northern hemisphere gridpoints). Unless the authors have a compelling reason to focus on the domain north of 65N (and also to exclude Baffin/Hudson Bay), I suggest using a Northern Hemisphere domain throughout the paper. This would greatly reduce confusion and make the results more plainly interpretable. This would also make these results directly comparable to the seasonal prediction skill estimates that the authors cite on 2.7, which would make this work much more relevant

to a broader community.

The authors' definition of Arctic domain and corresponding SIE (SIE_AO in the manuscript) is confusing because it systematically excludes many regions of high winter SIE variability, including the Labrador Sea, Bering Sea, Sea of Okhotsk, and Hudson Bay. This means that SIE_AO behaves like pan-Arctic SIE during the summer months, and behaves like GIN and Barents SIE in the winter months. In the melt/growth seasons, SIE_AO is a complex mix between these two. For each month, the reader is forced to perform a mental masking of the Arctic and think about what regions are actually contributing to SIE_AO variability in that given month. This significantly clouds the results of the paper. My specific comments related to this confusion are:

3.27-32: Figure 1a shows significantly higher melt season to growth season reemergence that Fig S1a. This is because Barents/GIN SIE anomalies are more persistent than anomalies in other Arctic regions, and these anomalies dominate the winter SIE_AO signal. I suggest checking the ratio of March SIE_AO standard deviation to pan-Arctic SIE standard deviation. This will indicate the amount of variance being lost due to the chosen AO mask (more on this in Major Comment 3, below)

4.4: The RMSE values in Fig. 2b are artificially low because SIE_AO doesn't have much winter SIE variability.

4.4-9: Why are the ACC values in Fig 2a and Fig S2a so different? In Fig. S2a there are a number of cases in which the short lead forecasts are less skillful than the long lead forecasts. For example, for the Jan 1 initialization, the lead 0-2 skill is substantially lower than the lead 9-11 skill. This is strange behavior and should be reported/commented on. Fig S2a is highly relevant as a direct comparison with other hindcast studies. Therefore, I believe that this figure should be a centerpiece of this paper.

4.13-16: The difference between Fig 2d and Fig S2d directly shows the effect of the domain choice. I expect this effect to be even larger for Jan, Feb, Mar, Apr sea ice. On

the other hand, the September SIE curves in Fig 2c and Fig S2c are identical.

4.27-29: The summer to winter differences in SIV-SIE correlations are much less pronounced when using a northern hemisphere domain for SIE (Fig 3a vs Fig S3a). This should be commented on in the text. Also, in Fig. S3 is SIV/OHC computed north of 65N or using a northern hemisphere domain?

5.5-6: This is not very surprising, given that other most other regions have been excluded!

5.6-7: This may be true, but the domain choice biases results towards finding a signal in the Barents/GIN seas.

Major Comment 2) Definition of OHC and advection mechanism

The authors define ocean heat content by integrating vertically from the base of the mixed layer to 200m depth. What is the rationale for excluding the mixed-layer heat content from this integral? I believe it is crucial to include the heat content from the mixed layer, as this is the heat that has direct access to the sea ice and therefore has greatest potential to influence sea ice variability. Moreover, by excluding the mixed-layer heat content, the OHC field becomes undefined when mixed layers become deeper than 200m in the winter months. This creates a very notable "hole" in the winter OHC fields in the Barents and GIN Seas. The authors claim that shifting correlation patterns in Fig 4c-f are evidence of advective processes. However, the main feature that I see is a shifting domain over which the OHC field is defined.

I strongly suggest the authors recompute OHC by integrating from the surface to 200m, and produce new versions of Fig 3 and 4 using this OHC field. This will allow the maps in Fig 4c-f to be defined at all gridpoints, and allow for a better assessment of the proposed adjective mechanism. Also, I am interested to see if the winter OHC correlations in Fig 3d-f become stronger with this new definition.

Also, is the December SIE_AO time series used in Fig. 4 computed using the model-

predicted SIE or observed SIE? In other words, is this proposed mechanism based on correlations with observations, or is it a "perfect model" mechanism?

Major Comment 3) Lagged correlation analysis

The lagged correlation results shown in Fig. 1a are significantly higher than those reported in Day et al. (2014). On first reading, this seems like a striking discrepancy. However, I believe this difference can primarily be attributed to the authors SIE_AO domain choice. It needs to be made very clear that Fig. 1a should not be compared directly with the Day et al (2014) results. Also, SIE_AO lagged correlations with NSIDC data should be added to Fig S1. Note that changing from the AO domain to the NH domain would alleviate this concern.

Minor Comments:

1.29: I suggest changing "predictions" to "projections", to make this distinct from the seasonal predictions that are the primary focus of this paper.

2.6: Is this based on detrended SIE or full SIE anomalies?

3.1: Should specify that this is ocean temperature.

3.2: What ocean data goes into the objective analysis of Ishii et al. (2006)? What SIC data is used?

3.19-20: This is unclear and needs to be explained more precisely.

3.28: How close is the SIC from Ishii et al. (2006) to SIC observations? Are there any known biases/differences?

Fig 2: Legends should be added to panels c and d

Fig 2 caption: Is July 1 referring to panel c and Jan 1 referring to panel d? This is currently unclear.

4.18-20: I disagree with the second half of this sentence. The July 1 forecasts appear

to have significant skill for Oct, Dec, Feb, and Mar.

4.19: What is "the longest lead time" referring to here? Do you mean "long lead times"?

Figure 4: Text labels should be added to the various panels to make this figure more readable.

5.24-26: I suggest adding Fig. S7 to the manuscript. Also, in this figure is the September SIE_AO the observed time series, or the time series from the hindcast experiments? This needs to be clarified.

6.7-9: These two sentences contradict one another. Please clarify.
* * *

---

## Referee Comment (RC2) · Anonymous Referee #2 · 7 Sep 2017

**Summary**

The authors present results on Arctic sea-ice extent prediction skill obtained with a MIROC-based forecast system. Further, they explore possible reasons for differences in skill in different times of the year based on lagged correlation and regression patterns, focussing on preceeding states of the (subsurface) ocean heat content and of the sea-ice itself.

In general, The paper is generally well-written and provides interesting results that merit publication. However, there are some points that in my view need further scrutiny. For example, the conclusion that the advection of subsurface water masses from the

[Figure]

Altantic Ocean into the Barents Sea, though plausible, is in my view not sufficiently supported by the results shown. Also, the definition of the subsurface ocean heat content and how it's interpreted deserves additional attention, and the rationale behind performing the lagged correlation/regression analysis primarily based on the hindcasts rather than on the control run, and what might cause differences between them, needs clarification. In addition, there is quite a number of minor issues, listed below.

Therefore I recommend the manuscript should be reconsidered after major revisions.

**Specific comments**

P1L8: The term "seasonal-to-interannual" should be shifted in front of "predictions".

P1L10: "of up three years" - here is a word ("to") missing.

P1L12: "December SIE_AO can be predicted up to 1 year ahead" - I suggest that this statement should be made more quantitative, e.g., by providing the ACC, and maybe also substantiated with the corresponding p-value.

P1L13-15: The role of advection as indicated here is in my view insufficiently supported by the results shown; see details below.

P1L23: "problem" - just as a side remark, I think this judgmental term adds an unnecessary political dimension to this observation.

P2L1: "or" - I think I know what is meant, but using "or" here seems illogical.

P2L2: "the potential predictability for sea ice extent is continuously one to two years" - I think this statement again needs some numbers; theoretically, marginal (but pratically meaningless) potential predictability should be out there for very long lead times, whereas pratically meaningful potential predictability survives much shorter lead times. At least, something like an ACC threshold which is considered to distinguish "meaningful" from "no" skill should be provided. (Note that "statistical significance" is not necessarily the correct concept needed here.)

P2L5-6: "The observed Arctic sea ice extent based on ensemble hindcasts can be predicted up to 2–7 and 5–11 months ahead for summer and winter" - see my previous remark.

P2L16: Again, I think that the term "seasonal-to-interannual" needs to be relocated, this time in front of "predictability".

P3L7-8: "eight ensemble members produced by perturbing the sea surface temperature based on the observational errors" - I am wondering whether these perturbations are able to generate any meaningful spread, given that the 3D ocean and atmosphere are assimilated towards the same, gap-free, reanalyses. Or, are the differences just very small (and all "assimilations" thereby very similar; note that Fig.2 also shows just one single "assimilation"), but of course sufficient to trigger subsequent divergence during the free forecast/hindcast runs due to atmospheric chaos, so that the same effect could have been obtained with quasi arbitrary small initial perturbations? Maybe the authors can comment.

P3L17-18: "the detrended components were calculated by subtracting monthly linear trends during 1980–2009 from the original monthly data, and anomalies are defined as deviations from the climatology from 1980–2009" - are not the "detrended components" mentioned at the beginning of this sentence already the "anomalies"?

P4L6-7: "September SIE_AO can be dynamically predicted from the previous July" - again, I think this statements needs some quantification; the same holds for the subsequent sentence.

P4L8-9: "The ACC is also significant for the winter SIE_AO, in particular for December, except for the hindcasts started from April 1st, indicating the potential use of dynamical forecasts up to 1 year ahead" - the fact that December SIE_AO is more skillfully predicted by the January hindcasts than by the April hindcasts, also visible in Fig.2, deserves more explanation. While such "reemergence" of skill is often encountered when simple statistical relations - like persistence - are used, in situations with strong

seasonal cycles like given for sea ice, to my understanding this is not to be expected for dynamical forecasts: the closer to the target date they are initialised (taking into account current as well as past observations!), the better should the dynamical forecasts become. To be specific, the OHC content anomalies put into the January hindcasts should also make it into the April hindcasts, although subject to some advection etc. Instead, could this unexpected drop of forecast skill be a mere matter of sampling uncertainty?

P4L10: "The RMSE for all 10 hindcasts increases throughout the melting and early freezing seasons (July–October), before decreasing in November–June" - to be precise, it appears that the RMSE does not "increase" and "decrease" during those periods, but that it "is larger" and "is smaller" (with the change happening inbetween).

P4L18-19: See again my comment on P4L8-9!

P4L19: "those started from July 1st, in which only the September SIE_AO is significant" - this statement seems to contradict Fig.2 where there are many "significance stipples" for other target months as well.

P4L22-23: "the SIV_AO is defined as the sum of the grid cell volumes obtained by multiplying the sea ice thickness (SIT) by the SIC for grid cells with SIC greater than 15 %" - if I am not mistaken, the multiplication by the grid-cell areas is missing here, no?

P4L23-25: "the OHC_AO is the vertically integrated temperature multiplied by the density and specific heat capacity of seawater from the mixed layer depth (MLD) to a depth of 200 m, in the area north of 65° N" - (i) The way it's defined here, temperature is vertically integrated instead of averaged, so the distance from the MLD to 200m directly enters the "OHC" and should thereby dominate variations in "OHC" instead of temperature variations, which seems odd. Please clarify. (ii) Why is not the same area used as for SIE_AO, that is, excluding Hudson Bay and Baffin Bay?

P5L7-18 and Fig.4c-f: I am not convinced that the "advection and emergence hypothesis" constructed here is sufficiently supported by the results shown. Firstly, some of the apparent propagation of ("subsurface") OHC anomalies from off the Scnadinavian western coast to the eastern Barents Sea might be simply due to a slight shift of the area with a mixed layer deeper than 200m (areas with quite deep convection): a larger part of the Barents Sea is thereby effectively "masked" in March compared to December in Fig.4. Secondly, the sea-ice edge extends further into the Barents Sea in March compared to December (I assume this is true also in these simulations), and ocean temperatures under ice are subject to weaker variability (with the surface being tied to the freezing point). Thirdly, the rather narrow stripe of anomalies off the Scandinavian coast in March - an important part of the presented explanation - is not present in the control run (Fig. S6). Maybe some clarification could be provided if Fig.S5 was also provided for lags -3, -6, and -9 months? It might also help to clarify things if the integration/averaging was done between fixed depths, so that nothing is masked and the MLD changes do not superimpose temperature anomaly changes. Even more simply, showing just SST anomalies might help.

P5L15-16: "The above features are also found in the control run, suggesting that the advection processes of the OHC in the hindcasts are not due to processes distorted by the influence of initialization or climate drift in MIROC5" - In fact, I do not quite understand the reason why the main figures related to the lagged correlation and regression alaysis are not based on the control run in the first place. Maybe it's just me, but I am somewhat confused why this should be done primarily for the hindcasts, where also the statistical sampling is much worse. If the main analysis was based on the control run, however, it would make sense to show corresponding results for the hindcasts as a supplement, to prove that the shown relations still hold, no?

P5L24: "the persistence of sea ice states initialized in July persists" - the first word maybe should be "anomalies" or similar?

P26-27: "possible mechanisms or sources cannot be detected in the hindcasts started from April 1st (Fig. S8)" - I'd like to repeat my points that this might be partly due

to sampling, and that important regions are "masked" due to the MLD-related OHC definition. I would argue that Fig.S6d, based on the control run (implying better sampling, although showing March instead of April), supports the notion that the April state should be at least as informative as the January state to predict September SIE_AO.

P6L4-6: "Numerical experiments to confirm whether the subsurface OHC anomalies 5 originating from the North Atlantic control the December sea ice extent in the BS and eventually in the Arctic Ocean will be explored in future work." - I am actually quite curious to see results of such interesting experiments!

P6L7-9: The first two sentences of this paragraph seem to contradict each other.

P6L20: "Further improvements in the predictability of sea ice" - here I would recommend to avoid the term "predictability (of)" because in my view "skill to predict" is more accurate.

Fig.2: I would find it helpful if the situations shown in panels c) and d) could be highlighted in panels a) and b), e.g., by black boxes around the corresponding fields of the heat maps. Also, do I understand correctly that panel c) corresponds to a 3 months lead time, whereas panel d) corresponds to a 11 months lead time? That could be stated more clearly in the caption.

---

## Author Comment (AC1) · 9 Oct 2017

Manuscript tc-2017-122

Mechanisms influencing seasonal-to-interannual prediction skill of sea ice extent in the Arctic Ocean in MIROC

Jun Ono, Hiroaki Tatebe, Yoshiki Komuro, Masato I. Nodzu, and Masayoshi Ishii

October 9, 2017

**Response to Anonymous Referee #1**

We deeply appreciate the referee's kind remarks about our paper. Detailed comments from referee are numbered consecutively and cited in italics, followed by our reply in bold face.

This work investigates the seasonal-to-interannual prediction skill of Arctic sea-ice extent (SIE) using a set of hindcast experiments performed with the MIROC GCM. The authors investigate prediction skill for detrended Arctic SIE, identifying skillful predictions up to one year in advance. They also examine the key physical mechanisms impacting prediction skill, concluding that North Atlantic ocean heat content anomalies are a source of skill for December SIE predictions.

I commend the authors for their focus on physical mechanisms and their relation to the reported SIE prediction skill. However, I have a number of serious concerns with the manuscript in its present form. In particular, my major concerns are: (1) the authors' choice of Arctic domain, and how this choice biases and confuses results throughout the manuscript; (2) the definition of ocean heat content and its impact on the proposed advective ocean heat content mechanism; and (3) the apparent disagreement of SIE lagged correlation values with previously published literature. Specific comments detailing these concerns are provided below.

Thank you very much for your concerns on our study. (1) Since we focus on the physical processes in the Arctic Ocean, we did not change the domain (please read our responses to referee's comments 1 and 2). (2) We recalculated ocean heat content and newly reconstructed Figures 3 and 4, according to your suggestions, and also partly rewrote the text. (3) Since our statements in the previous manuscript were not correct, we rewrote the text.

Note: I will use the convention p.l throughout this review to refer to page number p and line number l of the discussion paper.

**Major Comments:**

Before beginning the major comments, I would like to clarify a convention. The authors use a different lead-naming convention than the hindcast studies cited on 2.7. For example, a July 1 forecast of September SIE is referred to as a "lead-2" forecast in the literature cited on 2.7. In the manuscript, the authors refer to this forecast as a "lead-3" forecast. The authors should change their naming convention to be consistent with previous hindcast studies. I will use the commonly used convention in this review.

Thank you very much for letting us know about a lead-naming convention. In accordance with your advice, we modified the lead-naming and the corresponding text. For example, we replaced "1 year" with "11 months" in the revised manuscript (1.12).

**Major Comment 1) Choice of Arctic domain**

1. The author's define their Arctic Ocean domain as all gridpoints north of 65N. They also exclude Baffin Bay and Hudson Bay from their Arctic Ocean domain without providing any justification for this decision. The Arctic Ocean domain choice directly affects the interpretation of essentially all reported results in the paper. I suspect that Figures 1, 2, 3, and 4 would all be notably different if the authors analyzed the commonly used pan-Arctic domain (i.e. all northern hemisphere gridpoints). Unless the authors have a compelling reason to focus on the domain north of 65N (and also to exclude Baffin/Hudson Bay), I suggest using a Northern Hemisphere domain throughout the paper. This would greatly reduce confusion and make the results more plainly interpretable. This would also make these results directly comparable to the seasonal prediction skill estimates that the authors cite on 2.7, which would make this work much more relevant to a broader community.

So far, many previous studies on the predictability of Arctic sea ice extent with climate model have focused on the Pan-Arctic (or the Northern Hemisphere) domain. Furthermore, recent studies (1Sigmond et al., 2016; 2Bushuk et al., 2017) have evaluated the regional predictability in the Pan-Arctic domain. On the other hand, we focus on physical processes in the Arctic Ocean interior contributing to the seasonal-to-interannual predictability of the Arctic sea ice extent. In the present study, therefore, we would like to use the domain north of 65°N where sea ice has experienced rapid changes especially in the Pacific Sector of the Arctic Ocean (e.g., 3Comiso, 2012). In that case, the Baffin Bay and Hudson Bay are partly included in the domain, but the directions of main surface currents are heading from the Arctic Ocean interior (shelves and basins) to the Baffin Bay through the straits of the Canadian Archipelago (e.g., 4Aksenov et al., 2011). Thus, direct impacts of the Baffin Bay and Hudson Bay and Hudson Bay on physical processes through the Arctic Ocean interior are considered to be small. To clearly extract the impacts of physical processes through the Arctic Ocean interior on the Arctic sea ice, we did not consider the Hudson Bay and Baffin Bay.

1. Sigmond, M., Reader, M. C., Flato, G. M., Merryfield, W. J., and Tivy, A.: Skillful seasonal forecast of Arctic sea ice retreat and advance dates in a dynamical forecast system, Geophys. Res. Lett., 43, 12457-12465, doi:10.1002/2016GL071396, 2016.

 Bushuk, M., Msadek, R., Winton, M., Vecchi, G. A., Gudgel, R., Rosati, A., and Yang, X.: Skillful regional prediction of Arctic sea ice on seasonal timescales, Geophys. Res. Lett., 44, doi:10.1002/2017GL073155, 2017.

3. Comiso, J. C.: Large decadal decline of the Arctic multiyear ice cover, J. Clim., 25, 1176-1193, 2012.

4. Aksenov, Y. Ivanov, V. V., A. J. G. Nurser, S. Bacon, I. V. Polyakov, A. C. Coward, A. C. N. Garaboto, and Moeller, A. B.: The Arctic circumpolar boundary current, J. Geophys. Res., 116, C09017, doi:10.1029/2010JC006637, 2011.

In the revised manuscript, we removed "Note that Hudson Bay and Baffin Bay are excluded" (3.23-24 in the previous manuscript) from the text, and newly added "In that case, the Baffin Bay and Hudson Bay are partly included in the domain, but the directions of main currents are heading from the Arctic Ocean interior (shelves and basins) to the Baffin Bay through the straits of the Canadian Archipelago (e.g., Aksenov et al., 2011). Thus, direct impacts of the Baffin Bay and Hudson Bay and Hudson Bay on the Arctic Ocean interior are considered to be small." to the text (3.30-33). In addition, we removed Figures S3 and S4

in the previous supplement to focus on the physical processes in the domain north 65°N, although Figures S1 and S2 are remained to compare the previous studies.

2. The authors' definition of Arctic domain and corresponding SIE (SIE\_AO in the manuscript) is confusing because it systematically excludes many regions of high winter SIE variability, including the Labrador Sea, Bering Sea, Sea of Okhotsk, and Hudson Bay. This means that SIE\_AO behaves like pan-Arctic SIE during the summer months, and behaves like GIN and Barents SIE in the winter months. In the melt/growth seasons, SIE\_AO is a complex mix between these two. For each month, the reader is forced to perform a mental masking of the Arctic and think about what regions are actually contributing to SIE\_AO variability in that given month. This significantly clouds the results of the paper. My specific comments related to this confusion are:

For the reasons mentioned in our response to referee's comment 1, the area north of  $65^{\circ}$ N excluding Baffin Bay and Hudson Bay is defined as the Arctic domain in this study. As you pointed out, since the Labrador Sea, Bering Sea, Sea of Okhotsk, and Hudson Bay are excluded, the signal of winter SIEAO might be limited to the Barents Sea and GIN Sea. However, one of the main results of this study is the December SIEAO. In that case, positive regression and correlation spatial patterns are seen in the Barents Sea even in the results for the Northern Hemisphere domain (please see Figure S4 in the previous supplement). Thus, the definition of the Arctic domain does not seem to affect the main results of this study, at least, for the December SIEAO.

3. 3.27-32: Figure 1a shows significantly higher melt season to growth season reemergence that Fig S1a. This is because Barents/GIN SIE anomalies are more persistent than anomalies in other Arctic regions, and these anomalies dominate the winter SIE\_AO signal. I suggest checking the ratio of March SIE\_AO standard deviation to pan-Arctic SIE standard deviation. This will indicate the amount of variance being lost due to the chosen AO mask (more on this in Major Comment 3, below)

According to your advice, we checked the ratio of  $SIE_{AO}$  standard deviation to pan-Arctic SIE standard deviation for March. As a result, the value was 0.64. As you suggested, the remaining 36% is lost due to the domain selection, which might be explained by variability in the Labrador Sea, Bering Sea, Sea of Okhotsk, and Hudson Bay, and affect the

difference in the winter reemergence between Figure 1a and Figure S1. In the revised manuscript, we removed "In addition, the correlation coefficients are higher than those shown in Day et al. (2014b), for example, at a lead time of one month for May. This may be due to differences in the observations, temporal periods, and areas used for calculating the sea ice extent (Fig. S1)" (3.30-32 in the previous manuscript) from the text, and then added "As for the SIE in the Northern Hemisphere (Fig. S1a), the correlation patterns are similar those in Day et al. (2014b), except for a lead time of one month for May which may be due to difference in observations (Fig. S1d). However, reemergence in winter is weaker than that for SIEAO. This is because SIEAO exclude other regions contributing to the winter sea ice variability." to the text (4.9-12).

4. 4.4: The RMSE values in Fig. 2b are artificially low because SIE\_AO doesn't have much winter SIE variability.

Referee is quite correct. We added the reason why the RMSE values are low in winter as follows. "The RMSE values in winter are large (Fig. S2b) compared to Fig. 2b because  $SIE_{AO}$  does not include the area where sea ice variability is large." to the text (4.28-29).

5. 4.4-9: Why are the ACC values in Fig 2a and Fig S2a so different? In Fig. S2a there are a number of cases in which the short lead forecasts are less skillful than the long lead forecasts. For example, for the Jan 1 initialization, the lead 0-2 skill is substantially lower than the lead 9-11 skill. This is strange behavior and should be reported/commented on. Fig S2a is highly relevant as a direct comparison with other hindcast studies. Therefore, I believe that this figure should be a centerpiece of this paper.

In the Sea of Okhotsk, the Bering Sea, and the Labrador Sea, the ACC and RMSE between the observations and the hindcasts for sea ice concentration are lower and higher at the short lead time, respectively, for the hindcasts started in January and April 1st (not shown). This might influence the ACC for SIE in the Northern Hemisphere. In the revised manuscript, we added "The lower ACC at the short lead time for the hindcasts started from January and April (Fig. S2a) may be due to the lower ACC and higher RMSE for sea ice concentration in the Sea of Okhotsk, the Bering Sea, and the Labrador Sea (not shown)." to the text (4.26-28).

6. 4.13-16: The difference between Fig 2d and Fig S2d directly shows the effect of the domain choice. I expect this effect to be even larger for Jan, Feb, Mar, Apr sea ice. On the other hand, the September SIE curves in Fig 2c and Fig S2c are identical.

As you pointed out, the difference between Figure 2d and Figure S2d is due to the effect of the domain choice. In the revised manuscript, we added "The difference between Figure 2d and Figure S2d is also due to the effect of the domain choice." to the text (4.29-30).

7. 4.27-29: The summer to winter differences in SIV-SIE correlations are much less pronounced when using a northern hemisphere domain for SIE (Fig 3a vs Fig S3a). This should be commented on in the text. Also, in Fig. S3 is SIV/OHC computed north of 65N or using a northern hemisphere domain?

As you pointed out, correlation coefficients between SIV and SIE are significant in all season for the Northern Hemisphere domain (Figure S3a in the previous supplement). In the previous manuscript, we used the domain north of  $65^{\circ}$ N for computations of SIV and OHC. However, the same domain as the SIEAO should be used, as pointed out by referee #2. The difference between Figure 3a and Figure S3a might be due to the calculation method. In the revised manuscript, we recalculated SIV and OHC in the domain north of  $65^{\circ}$ N excluding the Hudson Bay and Baffin Bay (please see new Figure 3). Here OHC is integrated from the surface to a depth of 200 m, according to referee's comment 11. On the other hand, we removed Figure S3 in the previous supplement for the reasons mentioned in our response to referee's comment 1.

8. 5.5-6: This is not very surprising, given that other most other regions have been excluded!

As you pointed out, the sentence of "The most significant signals for both SIC and SIT are found in the Barents Sea (BS) of the Arctic Ocean (Figs. 4a and 4b)" may be not surprising result. However, even in the SIE in the Northern Hemisphere (Figures S4a and S4b in the previous supplement), similar but somewhat weak spatial patterns are seen in the BS. This indicates that the BS is one of dominant regions for the December SIE variability not only in the north of 65°N but also in the Northern Hemisphere.

**9. 5.6-7: This may be true, but the domain choice biases results towards finding a signal in the Barents/GIN seas.**

As mentioned in our response to referee's comment 8, significant signal in the BS can be seen even in the case of the Northern Hemisphere domain, although a signal in the GIN Sea disappears and significant signal appear partly in the North Pole, the Labrador Sea, and the Hudson Bay.

**Major Comment 2) Definition of OHC and advection mechanism**

10. The authors define ocean heat content by integrating vertically from the base of the mixed layer to 200m depth. What is the rationale for excluding the mixed-layer heat content from this integral? I believe it is crucial to include the heat content from the mixed layer, as this is the heat that has direct access to the sea ice and therefore has greatest potential to influence sea ice variability. Moreover, by excluding the mixed-layer heat content, the OHC field becomes undefined when mixed layers become deeper than 200m in the winter months. This creates a very notable "hole" in the winter OHC fields in the Barents and GIN Seas. The authors claim that shifting correlation patterns in Fig 4c-f are evidence of advective processes. However, the main feature that I see is a shifting domain over which the OHC field is defined.

As suggested by the previous studies [e.g., 5Nakanowatari et al., 2014], ocean temperatures around a depth of 200 m are effective for the sea ice prediction at the long lead-time. Motivated by the previous studies, we focused on the subsurface water as one of key variables that could provide memory on seasonal-to-interannual sea ice variability. In the previous manuscript, we did not consider the heat content within the mixed layer, to remove the direct effects due to the atmospheric heating and cooling. However, referee #2 has also commented the definition of the OHC and advection processes. In the revised manuscript, we recalculated the OHC. Please read our response to referee's comment 11.

5. Nakanowatari, T., Sato, K., and Inoue, J.: Predictability of the Barents sea ice in early winter: Remote effects of oceanic and atmospheric thermal conditions from the North Atlantic, J. Clim., 27, 8884-8901, doi:10.1175/JCLI-D-14-00125.1, 2014.

11. I strongly suggest the authors recompute OHC by integrating from the surface to 200m, and produce new versions of Fig 3 and 4 using this OHC field. This will allow the maps in Fig 4c-f to be defined at all gridpoints, and allow for a better assessment of the proposed adjective mechanism. Also, I am interested to see if the winter OHC correlations in Fig 3d-f become stronger with this new definition.

According to your suggestions, we reconstructed Figures 3 and 4 using the OHC from the surface to a depth of 200 m (please see new Figures 3 and 4), and rewrote the text (please read Section 4 in the revised manuscript). For comparison, we also added Figures 3d-3f and Figures 4c-4f in the previous manuscript to supplement as new Figure S4.

12. Also, is the December SIE\_AO time series used in Fig. 4 computed using the model-predicted SIE or observed SIE? In other words, is this proposed mechanism based on correlations with observations, or is it a "perfect model" mechanism?

In Figure 4, we used only data from the hindcasts (i.e., the model-predicted SIEAO).

Major Comment 3) Lagged correlation analysis

13. The lagged correlation results shown in Fig. 1a are significantly higher than those reported in Day et al. (2014). On first reading, this seems like a striking discrepancy. However, I believe this difference can primarily be attributed to the authors SIE\_AO domain choice. It needs to be made very clear that Fig. 1a should not be compared directly with the Day et al (2014) results.

As you pointed out, comparison of Figure 1a and result of Day et al. (2014) was not fair. In the revised manuscript, we rewrote the text by comparing Figure S1a and Day et al. (2014) as follows. "As for the SIE in the Northern Hemisphere (Fig. S1a), the correlation patterns are similar those in Day et al. (2014b), except for one month lead time of May which may be due to difference in observations (Figs. S1d)" (4.9-10).

14. Also, SIE\_AO lagged correlations with NSIDC data should be added to Fig S1. Note that changing from the AO domain to the NH domain would alleviate this concern.

As suggested, we added "Lagged correlations of SIEAO with NSIDC data" to new Figure S1 (please see new Figure S1e). For the reasons mentioned in our response to referee's comment 1, however, we mainly show results using the domain north of 65°N.

Minor Comments:

15. 1.29: I suggest changing "predictions" to "projections", to make this distinct from the seasonal predictions that are the primary focus of this paper.

As suggested, we replaced "predictions" with "projections" (1.29).

16. 2.6: Is this based on detrended SIE or full SIE anomalies?

This is based on detrended SIE. We added "detrended" to the text (2.6).

17. 3.1: Should specify that this is ocean temperature.

As suggested, we rewrote the text (3.1).

18. 3.2: What ocean data goes into the objective analysis of Ishii et al. (2006)? What SIC data is used?

Ocean data is based on the latest observational databases [the World Ocean Database (WOD05), World Ocean Atlas (WOA05), and Global Temperature Salinity Profile Program (GTSPP) provided by the U.S. National Oceanographic Data Center (NODC) and a SST analysis [Centennial in situ Observation Based Estimates of variability of SST and marine meteorological variables (COBE SST); 6Ishii et al. (2005); 7Hirahara et al. (2014)]. Also, SIC data is based on satellite observations from the Nimbus-5 Scanning Multichannel Microwave Radiometer (SMMR), the Special Sensor Microwave Imager (SSM/I), and the Special Sensor Microwave Imager/Sounder (SSMIS; 8Armstrong et al., 2012).

6. Ishii, M., Shouji, A., Sugimoto, S., and Matsumoto, T.: Objective analyses of SST and marine meteorological variables for the 20th century using ICOADS and the Kobe Collection. Int. J. Climatol., 25, 865-879, doi:10.1002/joc.1169, 2005.

7. Hirahara, S., Ishii, M., and Fukuda, Y.: Centennial-scale sea surface temperature analysis and its uncertainty. J. Climate, 27, 57-75, doi:10.1175/JCLI-D-12-00837.1, 2014.

8. Armstrong, R. L., Knowles, K. W., Brodzik, M. J., and Hardman, M. A.: DMSP SSM/I-SSMIS Pathfinder daily EASE-grid brightness temperatures, Jan 1987-Dec 2011. National Snow and Ice Data Center, CO, digital media. [Available online at http://nsidc.org/data/nsidc-0032.html.], 2012.

In the revised version, we added "Ocean data is based on the latest observational databases [the World Ocean Database (WOD05), World Ocean Atlas (WOA05), and Global Temperature Salinity Profile Program (GTSPP) provided by the U.S. National Oceanographic Data Center (NODC) and a SST analysis [Centennial in situ Observation Based Estimates of variability of SST and marine meteorological variables (COBE SST); Ishii et al. (2005); Hirahara et al. (2014)]. Also, SIC data is based on satellite observations from the Nimbus-5 Scanning Multichannel Microwave Radiometer (SMMR), the Special Sensor Microwave Imager (SSM/I), and the Special Sensor Microwave Imager/Sounder (SSMIS; Armstrong et al., 2012)." to the text (3.3-9).

**19. 3.19-20: This is unclear and needs to be explained more precisely.**

Probably, we are misleading referee's comment. Here, we calculated the climate drift following to method by INTERNATIONAL CLIVAR PROJECT OFFICE (ICPO, 2011) to remove the climate drift from the hindcasts.

20. 3.28: How close is the SIC from Ishii et al. (2006) to SIC observations? Are there any known biases/differences?

Figure A1 shows the differences between Ishii et al. (2006) and HadISST for summer (July-August-September) and winter (January-February-March) sea ice concentration (SIC). Here we used sea ice concentration from HadISST as observation because of the same horizontal resolution ( $1^{\circ}$  x  $1^{\circ}$ ). In summer, higher SIC (+10%) are seen in the Atlantic Sctor of the Arctic Ocean and lower SIC (-10%) in the Pacific Sector (Figure A1a). Although the biased SIC patterns in winter are similar to those in summer except for the Okhotsk Sea (Figure A1b), particularly higher SIC (+20%) are apparent in the

GIN Sea, Labrador Sea. However, these differences are smaller than standard deviation in SIC from the HadISST.

Figure A1. Differences between Ishii et al. (2006) and HadISST for summer (JAS; July-August-September) and winter (JFM; January-February-March) averaged sea ice concentration (SIC, %). Positive and negative values mean that SIC is higher and lower in Ishii et al. (2006) than HadISST.

**21. Fig 2: Legends should be added to panels c and d**

As suggested, we added legends to Figures 2c and 2d. Please see new Figure 2.

22. Fig 2 caption: Is July 1 referring to panel c and Jan 1 referring to panel d? This is currently unclear.

**We modified Figure 2 caption. Please see new Figure 2.**

23. 4.18-20: I disagree with the second half of this sentence. The July 1 forecasts appear to have significant skill for Oct, Dec, Feb, and Mar.

**Referee is quite right. We removed "only" from the text (5.3).**

24. 4.19: What is "the longest lead time" referring to here? Do you mean "long lead times"?

As you pointed out, "the longest lead time" means long lead times. In the revised manuscript, we replaced "the longest" with "long" (5.3).

25. Figure 4: Text labels should be added to the various panels to make this figure more readable.

As suggested, we reconstructed Figure 4. Please see new Figure 4.

26. 5.24-26: I suggest adding Fig. S7 to the manuscript. Also, in this figure is the September SIE\_AO the observed time series, or the time series from the hindcast experiments? This needs to be clarified.

According to your suggestion, we added Figure S7 in the previous supplement to the main text as new Figure 5 after the modification using OHC from the surface to 200 m. Please see new Figure 5. Also, this figure is based on the hindcasts as in Figures 3 and 4.

27. 6.7-9: These two sentences contradict one another. Please clarify.

As you pointed out, these two sentences were contradictory. In the revised manuscript, we removed the second sentence "Nevertheless, we note that the forecast skill of summer  $SIE_{AO}$  is not necessarily low, because the hindcasts initialized in January and April have significant skills for  $SIE_{AO}$  in August and September" (6.7-9 of the previous version) from the text.

---

## Author Comment (AC2) · 9 Oct 2017

Manuscript tc-2017-122

Mechanisms influencing seasonal-to-interannual prediction skill of sea ice extent in the Arctic Ocean in MIROC

Jun Ono, Hiroaki Tatebe, Yoshiki Komuro, Masato I. Nodzu, and Masayoshi Ishii

October 9, 2017

**Response to Anonymous Referee #2**

We deeply appreciate the reviewer's kind remarks about our paper. Detailed comments from reviewer are numbered consecutively and cited in italics, followed by our reply in bold face.

*### Summary ###*

*The authors present results on Arctic sea-ice extent prediction skill obtained with a MIROC-based forecast system. Further, they explore possible reasons for differences in skill in different times of the year based on lagged correlation and regression pat- terns, focussing on preceeding states of the (subsurface) ocean heat content and of the sea-ice itself.*

*In general, The paper is generally well-written and provides interesting results that merit publication. However, there are some points that in my view need further scrutiny. For example, the conclusion that the advection of subsurface water masses from the Altantic Ocean into the Barents Sea, though plausible, is in my view not sufficiently supported by the results shown. Also, the definition of the subsurface ocean heat content and how it's interpreted deserves additional attention, and the rationale behind performing the lagged correlation/regression analysis primarily based on the hindcasts rather than on the control run, and what might cause differences between them, needs clarification. In addition, there is quite a number of minor issues, listed below.*

*Therefore I recommend the manuscript should be reconsidered after major revisions.*

**Thank you very much for your summary comments and suggestions. We respond to specific comments as below.**

*### Specific comments ###*

*1. P1L8: The term "seasonal-to-interannual" should be shifted in front of "predictions".*

**As suggested, we corrected it (P1L8).**

*2. P1L10: "of up three years" - here is a word ("to") missing.*

**Thank you. We replaced "of up" with "up to" (P1L10).**

*3. P1L12: "December SIE_AO can be predicted up to 1 year ahead" - I suggest that this statement should be made more quantitative, e.g., by providing the ACC, and maybe also substantiated with the corresponding p-value.*

**As suggested, we added "(anomaly correlation coefficient is 0.42)" to the text (P1L13).**

*4. P1L13-15: The role of advection as indicated here is in my view insufficiently supported by the results shown; see details below.*

**Please see our response to the referee's comment 20.**

*5. P1L23: "problem" - just as a side remark, I think this judgmental term adds an unnecessary political dimension to this observation.*

**According to your advice, we removed "An even more serious problem is the decline in Arctic sea ice thickness (Kwok et al., 2009), which has decreased by around 65% from 1975 to 2012 (Lindsay and Schweiger, 2015)", and added "Moreover, Arctic sea ice thickness has decreased by around 65 % from 1975 to 2012 (Kwok et al., 2009; Lindsay and Schweiger, 2015)" to the text (P1L23-24).**

*6. P2L1: "or" - I think I know what is meant, but using "or" here seems illogical.*

**We replaced "two- or five-year" with "two and five years" (P2L1).**

*7. P2L2: "the potential predictability for sea ice extent is continuously one to two years" - I think this statement again needs some numbers; theoretically, marginal (but pratically meaningless) potential predictability should be out there for very long lead times, whereas*

*pratically meaningful potential predictability survives much shorter lead times. At least, something like an ACC threshold which is considered to distinguish "meaningful" from "no" skill should be provided. (Note that "statistical significance" is not necessarily the correct concept needed here.)*

**According to [1]Blanchard-Wrigglesworth et al. (2011), predictability is considered to be significant when the root mean square deviation of the ensemble of prediction experiments is less than that of the reference based on an F-test (for example, please see Figure 1 of Blanchard-Wrigglesworth et al. (2011)). However, the specific value that is considered to distinguish "meaningful" from "no" skill is not found in the paper. It might be overlooked, but we did not add any number to the text.**

**1. Blanchard-Wrigglesworth, E., Bitz, C. M., and Holland, M. M.: Influence of initial conditions and climate forcing on predicting Arctic sea ice, Geophys. Res. Lett., 38, L18503, doi:10.1029/2011GL048807, 2011.**

*8. P2L5-6: "The observed Arctic sea ice extent based on ensemble hindcasts can be predicted up to 2–7 and 5–11 months ahead for summer and winter" - see my previous remark.*

**As you pointed out, the specific value like an ACC threshold should be provided in the text. Predictability up to 2-7 and 5-11 months are based on the several results by previous studies (e.g., [2]Chevallier et al., 2013; [3]Sigmond et al., 2013; [4]Wang et al., 2013; [5]Msadek et al., 2014; [6]Peterson et al., 2015; [7]Guemas et al., 2016; [8]Sigmond et al., 2016). For example, Chevallier et al. (2013) have provided values for correlations and bootstrap test in Table 1. Also, in the study of Sigmond et al. (2016), forecast skill is considered to be significant when anomaly correlation coefficient exceeds to 0.296. However, such a value is not necessarily described in the previous all papers, although the assessment methods for forecast skill are described. Thus, we would like to avoid providing something like an ACC threshold to the text.**

**2. Chevallier, M., Salas-Mélia, D., Voldoire, A., and Déqué, M.: Seasonal forecasts of the Pan-Arctic sea ice extent using a GCM-based seasonal prediction system, J. Clim., 26, 6092-6104, doi:10.1175/JCLI-D-12-00612.1, 2013.**

3. Sigmond, M., Fyfe, J. C., Flato, G. M., Kharin, V. V., and Merryfield, W. J.: Seasonal forecast skill of Arctic sea ice area in a dynamical forecast system, Geophys. Res. Lett., 40, 529-534, doi:10.1002/grl.50129, 2013.

4. Wang, W., Chen, M., and Kumar, A.: Seasonal prediction of Arctic sea ice extent from a coupled dynamical forecast system, Mon. Weather Rev., 141, 1375-1394, doi:10.1175/MWR-D-12-00057.1, 2013.

5. Msadek, R., Vecchi, G. A., Winton, M., and Gudgel, R. G.: Importance of initial conditions in seasonal predictions of Arctic sea ice extent, Geophys. Res. Lett., 41, 5208-5215, doi:10.1002/2014GL060799, 2014.

6. Peterson, K. A., Arribas, A., Hewitt, H. T., Keen, A. B., Lea, D. J., and McLaren, A. J.: Assessing the forecast skill of Arctic sea ice extent in the GloSea4 seasonal prediction system, Clim Dyn., 44, 147-162, doi:10.1007/s00382-014-2190-9, 2015.

7. Guemas, V., Chevallier, M., Déqué, M., Bellprat, O., and Doblas-Reyes, F.: Impact of sea ice initialization on sea ice and atmosphere prediction skill on seasonal timescales, Geophys. Res. Lett., 43, 3889-3896, doi:10.1002/2015GL066626, 2016.

8. Sigmond, M., Reader, M. C., Flato, G. M., Merryfield, W. J., and Tivy, A.: Skillful seasonal forecast of Arctic sea ice retreat and advance dates in a dynamical forecast system, Geophys. Res. Lett., 43, 12457-12465, doi:10.1002/2016GL071396, 2016.

*9. P2L16: Again, I think that the term "seasonal-to-interannual" needs to be relocated, this time in front of "predictability".*

**As suggested, we correct (P2L17).**

*10. P3L7-8: "eight ensemble members produced by perturbing the sea surface temperature based on the observational errors" - I am wondering whether these perturbations are able to generate any meaningful spread, given that the 3D ocean and atmosphere are assimilated*

*towards the same, gap-free, reanalyses. Or, are the differences just very small (and all "assimilations" thereby very similar; note that Fig.2 also shows just one single "assimilation"), but of course sufficient to trigger subsequent divergence during the free forecast/hindcast runs due to atmospheric chaos, so that the same effect could have been obtained with quasi arbitrary small initial perturbations? Maybe the authors can comment.*

**Thank you for your comments. As for assimilation experiments, the ensemble spreads for detrended $SIE_{AO}$ are range from $10^2$ to $10^3$ km$^2$ (not shown) and therefore the time series of SIE for each member appear to a single curve. As you pointed out, the spread is very small and therefore the same effect could be obtained with small initial perturbations. However, in the present study, we have not conducted any hindcasts with small initial perturbations, for example, by the lagged averaged forecast (LAF; [9]Hoffman and Kalnay, 1983) method. Thus we cannot evaluate whether the initial SST perturbations are an effective method for producing the ensemble members or not, which will be remained as future works. At least, the time series of the ratio of ensemble spread for hindcasts to the corresponding RMSE indicates that the ensemble spread for hindcasts have values close to the RMSE (Figure B1), although are small for September. The initial SST perturbation methods seem to produce the meaningful spread to some extent.**

[Figure]

**Figure B1. Time series of the ratio of prediction ensemble spread to the RMSE for (a) September started in July 1st and (b) December started in January 1st.**

**9. Hoffman, R. N., and Kalnay, E.: Lagged average forecasting, an alternative to Monte**

**Carlo forecasting, Tellus, 35A, 100-118, doi:10.1111/j.1600-0870.1983.tb00189.x.**

*11. P3L17-18: "the detrended components were calculated by subtracting monthly linear trends during 1980–2009 from the original monthly data, and anomalies are defined as deviations from the climatology from 1980–2009" - are not the "detrended components" mentioned at the beginning of this sentence already the "anomalies"?*

**The "detrended components" are not anomalies. Firstly, anomalies are calculated by the definition described in the text, and then the linear trend is removed from anomalies.**

*12. P4L6-7: "September SIE_AO can be dynamically predicted from the previous July" - again, I think this statements needs some quantification; the same holds for the subsequent sentence.*

**As suggested, we added the values of ACC to the text (P4L16, P4L17, and P4L19).**

*13. P4L8-9: "The ACC is also significant for the winter SIE_AO, in particular for December, except for the hindcasts started from April 1st, indicating the potential use of dynamical forecasts up to 1 year ahead" - the fact that December SIE_AO is more skillfully predicted by the January hindcasts than by the April hindcasts, also visible in Fig.2, deserves more explanation. While such "reemergence" of skill is often encountered when simple statistical relations - like persistence - are used, in situations with strong seasonal cycles like given for sea ice, to my understanding this is not to be expected for dynamical forecasts: the closer to the target date they are initialised (taking into account current as well as past observations!), the better should the dynamical forecasts become. To be specific, the OHC content anomalies put into the January hindcasts should also make it into the April hindcasts, although subject to some advection etc. Instead, could this unexpected drop of forecast skill be a mere matter of sampling uncertainty?*

**In the present study, the December SIE$_{AO}$ can be predicted from January 1st but not from April 1st. To provide more explanation, here we considered two possibilities for the reasons. Firstly, we created the same figure as Figure 4 for the control experiment (Figure S3) and the April hindcasts (Figure S5). As in Figure 4, significant regression and correlation patterns appear in Figures S3 and S5. This suggests that the same physical**

mechanism occurs in the hindcasts started from April 1st. Thus the sampling uncertainty may not be the main reason for difference between the January hindcasts and the April hindcasts. Secondly, we compared the SIC RMSE between the observations and the hindcasts. In the Barents Sea contributing to the skill of the December $SIE_{AO}$, the SIC RMSE in April is larger in the April hindcasts than the January hindcasts (Figure B2). Possibly, the larger RMSE at 0 month lead time in the Barents Sea is the reason why the December $SIE_{AO}$ cannot be predicted by the April hindcasts. In the revised manuscript, we added "In contrast, the December $SIE_{AO}$ cannot be predicted from April 1st (Fig. 2a), although significant regression and correlation patterns appear in the results for the April hindcasts (Fig. S5). This may be because the RMSE for April SIC in the Barents Sea is larger in the April hindcasts than the January hindcasts (not shown)." to the text (P6L6-8).

[Figure]

**Figure B2. April RMSE for sea ice concentration (%) at (a) 3 months lead time from the January hindcasts (HIND.JAN) and (b) 0 month lead time from the April hindcasts (HIND.APR).**

*14. P4L10: "The RMSE for all 10 hindcasts increases throughout the melting and early freezing seasons (July–October), before decreasing in November–June" - to be precise, it appears that*

*the RMSE does not "increase" and "decrease" during those periods, but that it "is larger" and "is smaller" (with the change happening in between).*

**The sentence in the previous manuscript was not precise. As suggested, we rewrote this part as follows. "The RMSE for all hindcasts is larger throughout the melting and early freezing seasons (July-October), before smaller values in November-June." (P4L20-21).**

*15. P4L18-19: See again my comment on P4L8-9!*

**Please read our response to referee's comment 13.**

*16. P4L19: "those started from July 1st, in which only the September SIE_AO is significant" - this statement seems to contradict Fig.2 where there are many "significance stipples" for other target months as well.*

**Your comment is quite right. We removed "only" from this sentence (P5L3).**

*17. P4L22-23: "the SIV_AO is defined as the sum of the grid cell volumes obtained by multiplying the sea ice thickness (SIT) by the SIC for grid cells with SIC greater than 15 %" - if I am not mistaken, the multiplication by the grid-cell areas is missing here, no?*

**Thank you. As you pointed out, this sentence was not precise, but the SIV itself was correctly calculated. We added "and the area" to the text (P5L7).**

*18. P4L23-25: "the OHC_AO is the vertically integrated temperature multiplied by the density and specific heat capacity of seawater from the mixed layer depth (MLD) to a depth of 200 m, in the area north of 65° N" - (i) The way it's defined here, temperature is vertically integrated instead of averaged, so the distance from the MLD to 200m directly enters the "OHC" and should thereby dominate variations in "OHC" instead of temperature variations, which seems odd. Please clarify.*

**In the previous manuscript, we did not consider the heat content in the mixed layer, in order to remove the direct effects due to the atmospheric heating and cooling. However, as**

you pointed out, our previous definition of the OHC is affected by seasonal changes in the distance from the MLD to a depth 200 m (i.e., water volume). According to suggestions from referee #1 and referee #2, in the revised manuscript, we recalculated the OHC from the surface to a depth of 200 m and rewrote the text using new Figures 3 and 4. For comparison, we also added Figures 3d-3f and Figures 4c-4f in the previous manuscript to supplement as new Figure S4.

*19. (ii) Why is not the same area used as for SIE_AO, that is, excluding Hudson Bay and Baffin Bay?*

As you pointed out, we should calculate in the same region used as for $SIE_{AO}$. In the revised manuscript, we recalculated the SIV and OHC in the domain north of $65^o$N excluding Hudson Bay and Baffin Bay. Please see new Figure 3.

*20. P5L7-18 and Fig.4c-f: I am not convinced that the "advection and emergence hypothesis" constructed here is sufficiently supported by the results shown. Firstly, some of the apparent propagation of ("subsurface") OHC anomalies from off the Scnadinavian western coast to the eastern Barents Sea might be simply due to a slight shift of the area with a mixed layer deeper than 200m (areas with quite deep convection): a larger part of the Barents Sea is thereby effectively "masked" in March compared to December in Fig.4 Secondly, the sea-ice edge extends further into the Barents Sea in March compared to December (I assume this is true also in these simulations), and ocean temperatures under ice are subject to weaker variability (with the surface being tied to the freezing point). Thirdly, the rather narrow stripe of anomalies off the Scandinavian coast in March - an important part of the presented explanation - is not present in the control run (Fig. S6). Maybe some clarification could be provided if Fig.S5 was also provided for lags -3, -6, and -9 months? It might also help to clarify things if the integration/averaging was done between fixed depths, so that nothing is masked and the MLD changes do not superimpose temperature anomaly changes. Even more simply, showing just SST anomalies might help.*

Thank you for your suggestions. As you pointed out, ocean is masked when the mixed layer depth become deeper than a depth of 200 m. Firstly, we recalculated the OHC from the surface to a depth of 200 m in the revised manuscript, as mentioned in our response to referee's comment 18. Next we reconstructed new Figure 4 using new OHC and partly

**rewrote the Section 4 (P5L1-P6L21).**

*21. P5L15-16: "The above features are also found in the control run, suggesting that the advection processes of the OHC in the hindcasts are not due to processes distorted by the influence of initialization or climate drift in MIROC5" - In fact, I do not quite understand the reason why the main figures related to the lagged correlation and regression alaysis are not based on the control run in the first place. Maybe it's just me, but I am somewhat confused why this should be done primarily for the hindcasts, where also the statistical sampling is much worse. If the main analysis was based on the control run, however, it would make sense to show corresponding results for the hindcasts as a supplement, to prove that the shown relations still hold, no?*

**Referee comments may be correct. However, the main analysis using data from the hindcast experiments appear to be natural, as the first step, in order to investigate the physical processes contributing to the prediction skill of SIE$_{AO}$. Meanwhile, since the hindcast data may be influenced by climate drift or initialization, a control experiment without initialization and anthropogenic effects is complementally used to interpret the analyzed results.**

*22. P5L24: "the persistence of sea ice states initialized in July persists" - the first word maybe should be "anomalies" or similar?*

**Here we would like to state that initialized sea ice states persist until September. In the revised version, we changed "the persistence of sea ice states" to "the sea ice states". (P6L13).**

*23. P26-27: "possible mechanisms or sources cannot be detected in the hindcasts started from April 1st (Fig. S8)" - I'd like to repeat my points that this might be partly due to sampling, and that important regions are "masked" due to the MLD-related OHC definition. I would argue that Fig.S6d, based on the control run (implying better sampling, although showing March instead of April), supports the notion that the April state should be at least as informative as the January state to predict September SIE_AO.*

In the revised manuscript, we recalculated the OHC from the surface to a depth of 200 m, according to suggestions from referee #1 and referee #2, and then reconstructed Figure S8 in the previous supplement as new Figure S6. In the hindcasts started from April 1st, the September $SIE_{AO}$ shows similar lagged correlation patterns to the July hindcasts for $SIV_{AO}$ (Figure S6a) and $OHC_{AO}$ (Figure S6b). Thus, the same physical processes as the July hindcasts are expected to work in the April hindcasts. However, the positive regression and correlation patterns for SIC and SIT are weaker than those for the July hindcasts, particularly in the Pacific Sector of the Arctic Ocean (Figures S6c and S6d). In addition, the same figures based on the control experiment as Figure S6c and S6d are shown in Figure S7. Similar positive correlation and regression patterns for SIC and SIT clearly appear in the Pacific sector of the Arctic Ocean, as in Figure 5. As you pointed out, the sampling uncertainty may be one reason for unclear signals in the hindcasts started from April 1st.

In the revised manuscript, we removed "In contrast, possible mechanisms or sources cannot be detected in the hindcasts started from April 1st (Fig. S6), at least from the lagged correlation and regression analyses, although the September $SIE_{AO}$ is weakly correlated with the $SIV_{AO}$ and the $OHC_{AO}$." (P5L26-28 in the previous manuscript), and then newly added "In the hindcasts started from April 1st, the September $SIE_{AO}$ shows similar lagged correlation patterns to the July hindcasts for $SIV_{AO}$ (Fig. S6a) and $OHC_{AO}$ (Fig. S6b). Thus, the same physical processes as the July hindcasts are expected to work in the April hindcasts. However, the positive regression and correlation patterns for SIC and SIT are weaker than those for the July hindcasts, particularly in the Pacific Sector of the Arctic Ocean (Figs. S6c and S6d). In contrast, similar patterns to Fig. 5 clearly appear in the Pacific sector of the Arctic Ocean for the control experiment (Fig. S7). These results suggest that the persistence of sea ice contributes to the skill of September $SIE_{AO}$ started from April 1st, but the sampling uncertainty may lead to unclear signals in Fig. S6." to the text (P6L15-21).

*24. P6L4-6: "Numerical experiments to confirm whether the subsurface OHC anomalies 5 originating from the North Atlantic control the December sea ice extent in the BS and eventually in the Arctic Ocean will be explored in future work." - I am actually quite curious to see results of such interesting experiments!*

**Thank you for your interest. As mentioned in the text, we will conduct such an experiment in future works.**

*25. P6L7-9: The first two sentences of this paragraph seem to contradict each other.*

**As you and referee #1 pointed out, these two sentences were contradictory. We removed the second sentence "Nevertheless, we note that the forecast skill of summer SIE$_{AO}$ is not necessarily low, because the hindcasts initialized in January and April have significant skills for SIE$_{AO}$ in August and September" (P6L7-9 in the previous manuscript) from the revised text.**

*26. P6L20: "Further improvements in the predictability of sea ice" - here I would recommend to avoid the term "predictability (of)" because in my view "skill to predict" is more accurate.*

**As suggested, we replaced "predictability" with "skill to predict" (P7L14).**

*27. Fig.2: I would find it helpful if the situations shown in panels c) and d) could be highlighted in panels a) and b), e.g., by black boxes around the corresponding fields of the heat maps. Also, do I understand correctly that panel c) corresponds to a 3 months lead time, whereas panel d) corresponds to a 11 months lead time? That could be stated more clearly in the caption.*

**In the revised manuscript, we reconstructed Figure 2 following your suggestions. Please see new Figure 2.**

---

## Author Response (AR2)

Manuscript tc-2017-122

Mechanisms influencing seasonal-to-interannual prediction skill of sea ice extent in the Arctic Ocean in MIROC

Jun Ono, Hiroaki Tatebe, Yoshiki Komuro, Masato I. Nodzu, and Masayoshi Ishii

5 December 22, 2017

**Response to Anonymous Referee #1**

We deeply appreciate the referee's kind remarks about our paper. Detailed comments from referee are numbered consecutively and cited in italics, followed by our reply in bold face. We also requested native speakers of

10 English to proofread out English writing in the revised manuscript.

*Review of revised manuscript: Mechanisms influencing seasonal-to-interannual prediction skill of sea ice extent in the Arctic Ocean in MIROC by Ono et al.*

*This is a review of the revised manuscript. The authors have taken some effort to address my and the other review's comments. However, I still have two major concerns with the manuscript in its present form: (1) the*

15 *choice of Arctic domain and its influence on the results and conclusions of the study; and (2) the proposed advection ocean mechanism for winter prediction skill. These major concerns and some additional minor comments are outlined below.*

*Major Comments:*

*1. 1) Choice of Arctic domain*

20 *In my first review (RC1), I outlined a number of concerns directly related to the author's choice of using an Arctic domain defined as all gridpoints north of 65N. This choice creates confusion throughout the manuscripts, making the results difficult to interpret (see RC1, major comment 1). In my opinion, the authors' response to RC1 has not provided a compelling justification for this choice. If the authors insist on retaining this domain choice, the results throughout the manuscript need to be carefully caveated, so that readers do not misinterpret the*

25 *findings. Below are some changes that would need to be included if the choice of domain is retained (note that this list may not be exhaustive).*

**Thank you very much for your suggestions. Considering the difference in the Arctic domain, some sentences in the previous manuscript were not correct. In the revised manuscript, we partly rewrote the text according to referee's suggestions, but mainly showed the results for the domain north of 65$^{\circ}$N. Once**

30 **again, this study focuses on the physical processes in the Arctic Ocean interior because the ocean processes might contribute to the sea ice reduction in an area north of 65$^{\circ}$N (e.g., Polyakov et al., 2011[1]; 2012[2]). In that case, for example, the Sea of Okhotsk is not directly connected to the Arctic Ocean and the influences of Baffine Bay and Hudson Bay are thought to be small, as we mentioned in the text. Instead, the results for the Northern Hemisphere are shown in the revised supplement.**

**1. Polyakov, I. V., and Coauthors.: Fate of early 2000s Arctic warm water pulse, Bull. Amer. Meteor. Soc., 561-566, doi:10.1175/2010BAMS292.1, 2011.**

**2. Polyakov, I. V., Walsh, J. E., and Kwok, R.: Recent changes of arctic multiyear sea ice coverage and the likely causes, Bull. Amer. Meteor. Soc., 145-151, doi:10.1175/BAMS-D-11-00070.1, 2012.**

*2. 3.33: A line needs to added here explicitly stating that these results are not directly comparable with other hindcast studies. For example, "It should be noted that the results of this study are not directly comparable with other hindcast studies that focus on pan-Arctic SIE (e.g., Chevallier et al., 2013; Sigmond et al., 2013; Wang et al., 2013; Msadek et al., 2014; Peterson et al., 2015; Guemas et al., 2016; Sigmond et al., 2016), duo to this choice of Arctic Ocean domain."*

**As suggested, we added "Note that the results of this study are not directly comparable with other hindcast studies that focus on pan-Arctic SIE (e.g., Chevallier et al., 2013; Sigmond et al., 2013; Wang et al., 2013; Msadek et al., 2014; Peterson et al., 2015; Guemas et al., 2016; Sigmond et al., 2016), due to the choice of Arctic Ocean domain. " to the text (P4L3-6).**

*3. 4.11-12: I recommend explicitly noting the reason for this discrepancy: Winter SIE_AO variability is dominated by changes in the Barents and GIN Seas, which have long persistence timescales relative to other regions of winter ice variability.*

**As suggested, we replaced "This is because $SIE_{AO}$ exclude other regions contributing the winter sea ice variability." with "This is because the winter $SIE_{AO}$ variability is dominated by changes in the Barents and GIN Seas, which have long persistence timescales relative to other regions of winter sea ice variability." (P4L16-18).**

*4. 4.21-22: This sentence should be removed, as a comparison with Tietsche et al. (2014) is not appropriate given the different domains used in these studies.*

**As you pointed out, the sentence in the previous manuscript was not appropriate. In the revised manuscript, we removed the sentence.**

*5. 4.13-30: To alleviate issues with the choice of Arctic Ocean domain, the authors should consider using a normalized RMSE (NRMSE) metric, where the RMSE values are normalized by the standard deviation of each month. As it currently stands, these RMSE values are difficult to interpret giving the large seasonal cycle in RMSE. Put differently, it is hard to know from Fig. 2b what is a "good" or "bad" RMSE value.*

**Thank you for your suggestion. We reconstructed Fig. 2b following to your advice and rewrote the caption (Please see new Fig.2). We removed "The RMSE for all hindcasts is larger throughout the melting and early freezing seasons (July-October), before smaller values in November-June." in the previous manuscript and newly added the following sentences to the text.**

**"Here, the RMSE values are normalized by the standard deviation of each month." (P4L20-21).**

**"The RMSE values for the first several months are smaller than the standard deviation for all hindcasts."** (P4L26-27).

*6. 5.10-19: The authors have chosen to remove Fig. S3 of the previous supplementary material. This Figure should be added back into the supplement, as it provides important context for the results of the study. Also, a comment should be added to the text that summer-to-winter differences in SIV-SIE correlations are much less pronounced when using a northern hemisphere domain for SIE.*

**In Figure S3 in the supplement for open discussion, we had calculated SIV and OHC in the domain north of 65°N. This was not correct. In the revised version, we recalculated SIV and OHC for the Northern Hemisphere and reconstructed a new Fig. S3. Please see the revised supplement. In addition, please read the response to referee's comment 16.**

*7. 2) Advective ocean mechanism for winter prediction skill*

*The authors claim that Figs. 4c-f provide evidence for subsurface OHC anomalies advected from the North Atlantic. While plausible, this mechanism is not convincingly shown by Fig. 4. Aside from a small patch of positive correlation west of Novaya Zemlya in panels 4c and 4d, there is no clear evidence of OHC anomaly advection. Rather, the main signal appears to be a large stationary patch of OHC anomalies throughout the Barents and GIN Seas, which is present at lags of 9, 6, 3, and 0. The origin of these anomalies is not clear from Fig. 4.*

**Please read the response to referee's comment 8.**

*8. The authors claim that OHC anomalies "flow into the BS through advection." What is the evidence for this? Either more evidence needs to be provided or the statements about regarding the advection mechanism need to be appropriately qualified. For example, the advection hypothesis could be referred to as a "plausible mechanism."*

**Certainly, it might be difficult to claim, from Fig. 4c-4f, that OHC anomaly flow into the Barents Sea through the advection processes. As you pointed out, the signal mainly appears to be stationary, while partly appears to propagate to the eastern part of Barents Sea (in the western part of Novaya Zemlya). Considering that the Norwegian Atlantic Current tends to flow into the BS, it would be natural to think that the North Atlantic might be the source of OHC anomaly in the BS. In the revised manuscript, we rewrote the text according to suggestions from referee #1 and referee #2.**

**Firstly, we removed the sentences from P5L29 to P6L2 in the previous manuscript, as will be mentioned in the response to referee #2' comment 9. Secondly, we moved the sentences "In contrast … the January hindcasts (not shown)." (P6L6-8 in the previous manuscript) to the end of the third paragraph in section 4 (P5L32-P6L2). Thirdly, we rewrote the sentences "Hence, the OHC … December SIE$_{AO}$." (P6L3-12 in the previous manuscript) as follows. "Considering that the Norwegian Atlantic Current tends to flow into the BS (e.g., Polyakov et al., 2005), the North Atlantic might be the source of the OHC anomaly contributing to the significant skill of the December SIE$_{AO}$. A plausible mechanism is as follows: the OHC anomalies initialized in the North Atlantic flow into the BS through advection, subsequently emerge at the surface due to vertical mixing in winter, and affect the December sea ice distribution in the BS and eventually in the Arctic Ocean. This hypothesis is partly supported by Nakanowatari et al. (2014). As originally**

**proposed by Bushunk et al. (2017), our results suggest that the initialization of subsurface ocean temperature contributes to the skillful prediction of the winter sea ice extent in the BS." (P6L6-12).**

*9. 5.33-6.1: As mentioned in RC1 and RC2, this apparent advection from the North Atlantic is likely an artifact of only defining OHC below the mixed layer. When North Atlantic mixed layers deepen, this quantity becomes undefined, creating the undefined regions seen in Fig. S4. Therefore, this sentence should be removed.*

**As pointed out by referee #1 and #2, the sentences from P5L29 to P6L2 in the previous manuscript and Fig. S4 were not appropriate. As mentioned above, we removed them.**

*Minor comments:*

*10. 3.26-27: This statement remains unclear. How are the drifts removed? An additive correction? A regression based approach? Is the drift removal lead-time dependent? More detail is needed here.*

**In the revised version, we added "Here, the climate drift $T_{drf}$ is estimated as follows: $T_{drf}(\tau) = \frac{1}{N}\sum_{k=1}^{N}(T_p^k(\tau) - T_a^k(\tau))$, where $k = 1, \cdots, N$ is the initial time; $\tau$ is the forecast lead time; $T$ is the monthly quantity of interest, for example, the temperature and sea ice concentration; and the subscripts $p$ and $a$ represent the ensemble averaged prediction and the corresponding assimilation, respectively." to the text (P3L27-30).**

*11. 4.4: The autocorrelation coefficients are the skill of a persistence forecast. I recommend changing this line to: "We first examine the potential predictability of SIE_AO (Fig. 1), based on lagged auto-correlation coefficients, which is the skill of a persistence forecast."*

**Thank you. As suggested, we changed it (P4L9-10).**

*12. 4.9: Change to "…similar to those…"*

**As suggested, we changed it (P4L14).**

*13. 4.10: Change to "..due to differences in the observational time period."*

**As suggested, we changed it (P4L15-16).**

*14. 4.25: Change to "…mid-1980s and mid-1990s."*

**As suggested, we changed it (P4L29).**

*15. 5.14: Change to "…similar features…"*

**As suggested, we changed it (P5L15).**

*16. 5.17: I believe Fig. S3 here is referring to the old Fig. S3 (which is no longer in the supplement)*

**Thank you. You are quite right. As mentioned in our response to referee's comment 6, we added the same figure as Fig. 3 for the Northern Hemisphere to the revised supplement (Fig. S3), and we rewrote this part as follows.**

**"For the sea ice extent anomaly calculated in the Northern Hemisphere (Fig. S3), the patterns of the lagged correlation coefficients are broadly similar to those in Fig. 3. However, the correlations between the SIE and SIV are higher than those in the Arctic domain north of 65° N. One reason might be the contribution of sea ice variability south of 65° N. One reason might be the contribution of sea ice variability south of 65° N. In addition, the correlations between SIE and OHC show weak positive values from June to October in the hindcasts. This is partly because the OHC includes the regions where sea ice does not exist throughout the year." (P5L18-23).**

*17. 5.21: Change to "...model-predicted December SIE_AO"*

**As suggested, we changed it (P5L25).**

*18. 6.20: Rather than sampling uncertainty, I would suggest that a more likely explanation is errors in sea ice thickness initial conditions and model drift.*

**Based on your suggestion, we replaced "the sampling uncertainty" with "errors in the initial conditions for SIT and model drift" (P6L21).**

*19. 7.1-3: This sentence needs to be removed (the author's claimed to have removed it in their response, but seem to have forgot).*

**Thank you. You are quite right. We removed the sentence (P7L1-3 in the previous manuscript).**

Manuscript tc-2017-122

Mechanisms influencing seasonal-to-interannual prediction skill of sea ice extent in the Arctic Ocean in MIROC

Jun Ono, Hiroaki Tatebe, Yoshiki Komuro, Masato I. Nodzu, and Masayoshi Ishii

December 22, 2017

**Response to Anonymous Referee #2**

We deeply appreciate the referee's kind remarks about our paper. Detailed comments from referee are numbered consecutively and cited in italics, followed by our reply in bold face. We also requested native speakers of English to proofread out English writing in the revised manuscript.

*Overall the authors have done a good job at revising their manuscript. They have invested significant extra work in response to the reviewer comments, which I very much appreciate. In my view the results of the paper do merit publication, but I think that the points listed below still need to be addressed so that the results are not over-interpreted.*

*The numbers in the following refer to the numbering of author comments in the response of the authors.*

*###*

*Major points:*

*1. 20: The authors have followed the suggestion to compute OHC in a way so it better reflects actual temperature variations rather than shifts in the mixed-layer depth, which I think is very good. The resulting new Fig. 4 in my view confirms that the "advection story" is not really supported by the results. However, the authors now use the arguments that "the direct heating and cooling of atmosphere are considered to influence the above OHC through the sea surface" to explain why they base their further interpretation rather on the OHC according to the old definition integral between MLD and 200 m, which in my view is still questionable. Also, the statement "Correlation patterns between the SIE_AO and OHC [...] are not significant during the winter when the MLD is below a depth of 200 m" is in my view misleading: it suggests that statistical significance is involved, although the OHC values just become "undefined". Based on the old OHC definition, the authors state (first with some caution) that "the negative correlation and regression patterns appear to propagate from the North Atlantic to*

*the BS", but then (less cautiously and without any revision): "Hence, the OHC anomalies initialized in the North Atlantic flow into the BS through advection, subsequently emerge at the surface due to vertical mixing in winter, and affect the December sea ice distribution in the BS and eventually in the Arctic Ocean." Accordingly, the corresponding statement in the abstract still reads: "This skill is attributed to the subsurface ocean heat content*

5 *originating in the North Atlantic. The subsurface water flows into the Barents Sea from spring to fall and emerges at the surface in winter by vertical mixing, and eventually affects the sea ice variability there." I think this is an over-interpretation of the results.*

**Thank you very much for your suggestions. As mentioned in comments from referee #1 and referee #2, our**

10 **statements on advection processes may be over-interpretation. In the revised manuscript, we alleviated this part according to two referee's suggestions.**

**Firstly, we removed the sentences from P5L29 to P6L2 in the previous manuscript, as will be mentioned in the response to referee #2' comment 9. Secondly, we moved the sentences "In contrast … the January**

15 **hindcasts (not shown)." (P6L6-8 in the previous manuscript) to the end of the third paragraph in section 4 (P5L32-P6L2). Thirdly, we rewrote the sentences "Hence, the OHC … December SIE$_{AO}$." (P6L3-12 in the previous manuscript) as follows. "Considering that the Norwegian Atlantic Current tends to flow into the BS (e.g., Polyakov et al., 2005), the North Atlantic might be the source of the OHC anomaly contributing to the significant skill of the December SIE$_{AO}$. A plausible mechanism is as follows: the OHC anomalies**

20 **initialized in the North Atlantic flow into the BS through advection, subsequently emerge at the surface due to vertical mixing in winter, and affect the December sea ice distribution in the BS and eventually in the Arctic Ocean. This hypothesis is partly supported by Nakanowatari et al. (2014). As originally proposed by Bushunk et al. (2017), our results suggest that the initialization of subsurface ocean temperature contributes to the skillful prediction of the winter sea ice extent in the BS." (P6L6-12).**

**In the abstract, we changed "is" to "might be" (P1L13) and added "A plausible mechanism is as follows:" before "the subsurface …" (P1L14).**

*2. 13 (and 15, 23): I am not convinced by the offered explanation why December SIE_AO should be better*

30 *predictable from January than from April. It may well be related to the result that "the RMSE for April SIC in the Barents Sea is larger in the April hindcasts than the January hindcasts" (as shown in Fig. B2) - but this only*

*shifts the problem to the question why that should be so! As SIC is assimilated, the April SIC field should be much closer to observations right after initialization (at 0-month lag) than after an initialization in January (at 3 months lag), no? Is something going wrong with the assimilation/initialization? Maybe it's just me, but in my view it would be important to explain this properly.*

**Since the observed SIC is assimilated into the model with one-day interval in this study, as you pointed out, the April SIC field should be close to observation in the hindcasts started from April (0 month lag) compared to January (3 months lag). In the previous response to RC2, we showed the SIC RMSE in April for the January and April hindcasts. We think about the reason why the April SIC RMSE in the Barents**

10 **Sea is larger in the April hindcasts than January as follows: In this study, since we have not assimilated ocean data beneath sea ice into the model because of no reliable data, initialized ocean states underneath the sea ice are considered to be different from the real ocean. Particularly, in the Barents Sea where sea ice variability is related to the skillful prediction of December SIE$_{AO}$, standard deviation of sea ice is larger in April than January. In that case, initial shock might be large in April. This is one of the reasons for the**

15 **larger April SIC RMSE in the BS. To improve such a problem, reliable ocean data beneath the sea ice and the sophisticated assimilation method will be needed in future works.**

**In the revised manuscript, we added "In this study, since we do not assimilate ocean data beneath the sea ice, initialized ocean states underneath the sea ice are considered to be different from the real ocean.**

20 **Particularly, in the BS where sea ice variability is related to the skillful prediction of December SIE$_{AO}$, standard deviation of sea ice is larger in April than in January, and thus the initial shock might be large in April." to the text (P6L2-5).**

###

25 *Minor point:*

*3. 7 (and 8, 12): I would argue that in the referenced paper, as you say, statistical significance has been used to distinguish "significant predictability" from "no predictability", but in that case it's clear that "no predictability" implies just not sufficient signal to exceed the noise level given the sample size in that study. However, you*

30 *translate this into the statement "the study has shown that the potential predictability for sea ice extent is continuously one to two years" - which has lost the link to any quantitative threshold. If you'd rather say*

*something like "in that study, potential predictability remained statistically significant at lead times up to 1-2 years". I think that would be more sound.*

**Thank you. What we would like to say here is the latter sentence "in that study, potential predictability remained 
[revised manuscript text omitted]